



**Developing Functional Recharge Systems to Repel Saltwater**
**Intrusion via Integrating Physical, Numerical, and Decision-Making**
**Models for Coastal Aquifer Sustainability**
Yehia Miky[1], Usama Hamed Issa[2,3], Wael Elham Mahmod[3,4]
[1]Department of Geomatics, Faculty of Architecture and planning, King Abdulaziz University, Jeddah, Saudi Arabia;
yhhassan@kau.edu.sa
[2]Department of civil engineering, Faculty of Engineering, Minia University, Egypt; usama.issa@mu.edu.eg
[3]Department of civil engineering, college of Engineering, Taif University, Saudi Arabia; wemahmod@tu.edu.sa;
9 +966543656209
[4]Civil Engineering Department, Faculty of Engineering, Assiut University, 71515 Assiut, Egypt; wdpp2006@aun.edu.eg; +20-
11 01006328492,
orcid.org/0000-0003-4340-0525
*Correspondence to*: Wael Elham Mahmod (wdpp2006@aun.edu.eg)
**Abstract.** Controlling the hydraulic heads along the coastal aquifer may help to effectively manage saltwater intrusion,
improve the conventional barrier's countermeasure, and ensure the coastal aquifer's long-term viability. This study proposed a
framework that utilizes a decision-making model (DMM) by incorporating the results of two other models (physical and
numerical) to determine proper countermeasure components. The physical model is developed to analyze the behavior of
saltwater intrusion in unconfined coastal aquifers by conducting two experiments: one for the base case and one for the
traditional vertical barrier. MODFLOW is used to create a numerical model for the same aquifer, and experimental data is
used to calibrate and validate it. Three countermeasure combinations, including vertical barrier, surface, and subsurface
recharges, are numerically investigated using three model case categories. Category (a) model cases investigate the hydraulic
head's variation along the aquifer to determine the best recharge location. Under categories (b) and (c), the effects of surface
and subsurface recharges are studied separately or in conjunction with a vertical barrier. As a pre-set of the DMM, evaluation
and classification ratios are created from the physical and numerical models, respectively. The evaluation ratios are used to
characterize the model cases results, while the classification ratios are used to classify each model case as best or worst. An
analytic hierarchy process (AHP) as DMM is built using the classification ratios of hydraulic head (HHR), salt line (SLR),
intrusion (IR), repulsion (Rr), wedge area (WAR), and recharge (RER) as selection criteria to select the overall best model
case. The optimal recharging location, according to the results, is in the length ratio (LR) range from 0.45 to 0.55. Furthermore,
the DMM supports case3b (vertical barrier + surface recharge) as the best model case to use, with a support percentage of
47.93%, implying that this case has a good numerical model classification with a minimum IR of 67.9%, a maximum Rr of
29.4%, and an acceptable WAR of 1.25. The proposed framework could be used in various case studies under different
conditions to assist decision-makers in evaluating and controlling saltwater intrusion in coastal aquifers.
**Keywords:** Saltwater intrusion · Hydraulic heads · Unconfined coastal aquifer · Vertical barrier · Surface recharge · subsurface
recharge · Decision making · AHP
**1 Introduction**
Due to the natural effects of long-term climate change such as sea level change and tidal intensity fluctuations, seawater flows
toward the freshwater aquifers. In addition, increased water demands accompanied by anthropogenic activities such as
excessive pumping of freshwater in coastal areas cause the lowering of water tables as well as saltwater intrusion (Abd-Elaty
et al. 2019; Sutar and Rotte 2022). Saltwater intrusion lowers the potentiality and quality of freshwater in coastal regions, as
reported at many locations all over the world (Qi and Qiu 2011; Shi and Jiao 2014; Anders et al. 2014; Cary et al. 2015;


Srinivasamoorthy 2015; Abd-Elhamid 2016; Eissa et al. 2018; Abd-Elhamid et al. 2019; Pramada et al. 2021). Therefore, it is
important to control saltwater intrusion with efficient countermeasures to achieve sustainable freshwater sources.
Traditional methods for controlling saltwater intrusion include reducing pumping rates, relocating pumping wells, changing
pumping patterns, constructing physical subsurface barriers, and saltwater abstraction (Abd-Elhamid and Javadi 2011;
Kallioras et al. 2013; Cai et al. 2015;  Huang and Chiu 2018; Abd-Elhamid et al. 2019; Hussain et al. 2019). The limitations
and high costs of the aforementioned methods are substantial challenges to their implementation..
Artificial recharge techniques, such as surface and subsurface recharge systems, are critical for establishing hydraulic barriers
and mitigating the effects of saltwater intrusion. These techniques have several advantages compared to traditional methods,
including low cost, no inundation storage space, less water evaporation, and improved water quality. Although artificial
recharge has numerous advantages, it also has disadvantages, including groundwater contamination from surface water,
difficulty in implementation due to a lack of understanding of aquifer hydrogeological properties, the potential for
environmental damage and soil disturbance, and high maintenance costs. Surface recharge systems include ditches and
furrows, recharge basins, stream augmentation, and runoff conservation structures (terracing, contour bunds, percolation tanks,
gully plugs, Nalah bunds, and check dams) (Maliva 2020b, c; ASCE 2001). On the other hand, subsurface recharge systems
include subsurface injection wells, borewells, and recharging pits and shafts (Maliva 2020a, d; ASCE 2001). Combining
traditional and artificial recharge techniques is one way to overcome the disadvantages of both. Although many studies
investigate saltwater intrusion in coastal aquifers, only a limited number study the control methods of saltwater intrusion.
Physical and numerical models have not only proven to be more effective and economic tools for selecting the optimum
solutions for repelling saltwater intrusion but can be used to reduce the high cost of hydrogeological and environmental
investigations before constructing a full-scale project (Mantoglou 2003; Zhou, et al. 2003; Abarca et al. 2006; Singh 2015;
Abd-Elaty et al. 2019; Guo et al. 2019; M Armanuos et al. 2019).
Although physical and numerical models are effective economic tools for selecting the best solutions for repelling saltwater
intrusion, deficiencies in the acquisition of appropriate evidence to support the final decision are discovered. It is necessary to
use decision models in conjunction with physical and numerical models to guide stakeholders toward sustainable resource
management based on a set of criteria. The analytical hierarchy process (AHP) is a decision-making method that has been used
alone or in conjunction with other techniques such as GIS and fuzzy logic in a variety of groundwater-related fields. Based on
a broader set of criteria, this technique is used to guide stakeholders involved in groundwater development and sustainable
resource management (Vaidya and Kumar 2006; Alwetaishi et al. 2017). The applications of AHP in the field of groundwater
include assessing groundwater vulnerability by developing indices based on hydrogeological parameters and mapping
groundwater potential zones (Arunbose et al. 2021; Osiakwan et al. 2022; Ahmadi et al. 2021; Castillo et al. 2022; Achu et al.
2020; Sajil Kumar et al. 2022; Nithya et al. 2019; Phin et al., 2022; Zghibi et al., 2020; Mallick et al., 2019; Shao et al., 2020).
In the field of saltwater intrusion, a GIS-based AHP weighted index overlay analysis technique has been demonstrated to
determine the distribution of groundwater vulnerability (Gangadharan, Nila, et al. 2016; Güllü and Kavurmacı 2023). A fuzzy-
AHP evaluation model is developed for analyzing the level of seawater intrusion in long-term monitoring data from multiple
river basins (Yang et al., 2022). The AHP is also used to compute weights for the GALDIT parameters, which are used to
assess the vulnerability of coastal aquifers to saltwater intrusion (Pham et al., 2022).
According to the preceding overview, both traditional and artificial techniques of repelling seawater intrusion have limitations,
and using physical, numerical, and decision-making models is crucial. The unconfined coastal aquifer is investigated in this
work, and the surface and subsurface recharge methods, either alone or in conjunction with typical vertical barriers, are
analyzed by integrating physical, numerical, and decision-making models. On the other hand, the behaviors of saltwater
intrusion, groundwater flow, and hydraulic head are numerically investigated using three categories of model cases: category
(a), (b), and (c). The aims of this study are: (i) to examine experimentally the behavior of saltwater intrusion via coastal
unconfined aquifers with and without vertical barrier countermeasures; (ii) to develop a validated numerical model regarding


the experimental findings of transitory saltwater intrusion; (iii) to numerically analyze the flow behavior of saltwater and
freshwater, therefore identifying the saltwater-freshwater interaction via porous media; (iv) to identify the optimal recharging
location utilizing the location of the minimum hydraulic head; (v) to determine the optimal vertical barrier depth for saltwater
intrusion management; (vi) to identify the components of an effective countermeasure system, such as a vertical barrier, surface
recharge, and subsurface recharge, either alone or in combination; (vii) to develop a DMM model to aid decision makers in
the selection among several saltwater countermeasures and picking the most appropriate one depending on various demanding
scenarios.

## 2 Materials and Methodologies

Saltwater intrusion is investigated experimentally in part of this study by developing a laboratory physical model of an
unconfined coastal aquifer. Two experiments are carried out in this part, and ratios are formed using the dimension analysis
method, namely as evaluation ratios. These evaluation ratios are used to analyze and characterize the saltwater line and
hydraulic head variations of the numerical model cases, as forthcoming later. A numerical finite difference model is created,
and the validation and calibration processes are carried out using the experimental results. Following that, the numerical
repelling of saltwater intrusion is investigated, taking into account the combined effect of using vertical barriers with surface
or subsurface recharging systems, as depicted by model cases divided into three categories (a, b, and c) (seven cases in each
category). The results of the category (a) model cases reveal the location of the minimal hydraulic heads, which are expected
to be the locations of the indicated artificial recharge systems. A classification process is then implemented to classify model
cases in each category as best or worst model case using a developed set of ratios, namely classification ratios. Because each
model case is expected to have benefits and drawbacks, as well as several criteria governing the model cases, the benefits and
drawbacks of each model case should be quantified in order to identify the most effective one. Following that, the most
effective model case is decided on using a new DMM model based on the AHP technique. To make the final decision, two
selection levels (levels 1 and 2) are considered. **Figure 1** illustrates a flow chart for the framework of the study.





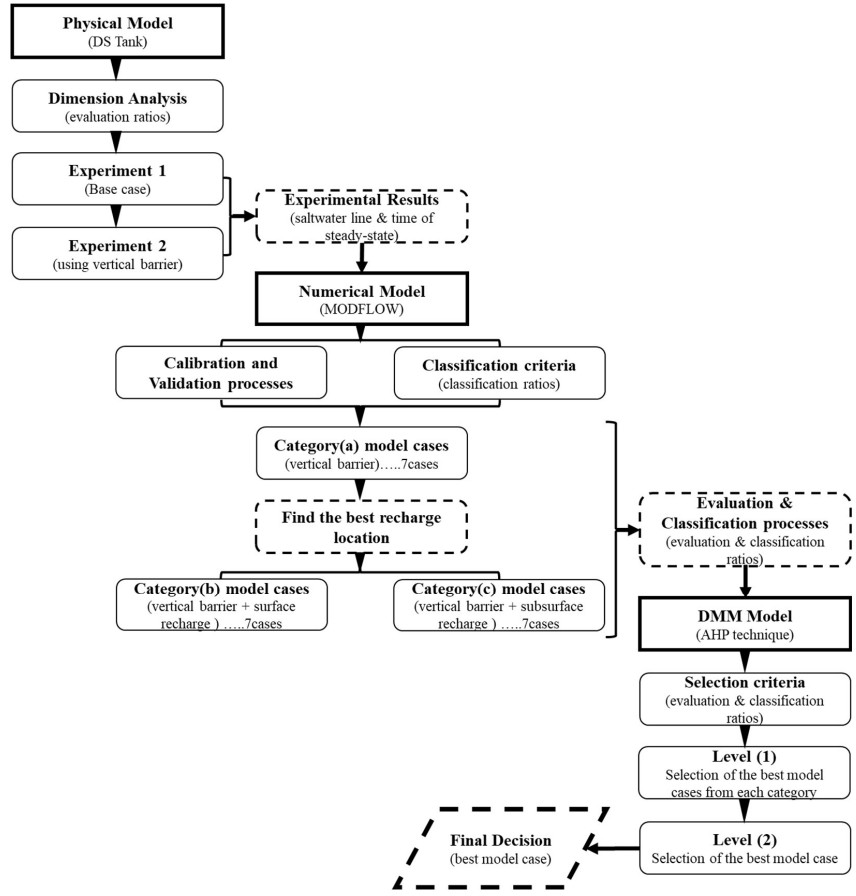


**Figure 1: Flow chart shows the proposed framework to identify most effective model case using the physical, numerical, and decision-making models**

**2.1 Experimental Setup**

**2.1.1 Drainage and Seepage Tank (DS Tank)**

The DS Tank is used in this study to visualize groundwater flow through permeable porous media. The model of the DS Tank that is used in the current study is HM 169 GUNT HAMBURG. The DS Tank consists of a porous media container, a lower water tank as a water source, a pump for the water flow, a valve to adjust the water supply, and measuring connections in the experiment section, which are connected to 14 glass tube manometers to display and measure hydraulic heads along the DS Tank. The sand container consists of an aluminum rectangular tank with a transparent front side (methacrylate material) to visualize groundwater flow and optimize observation of the experiments through the porous media. In the DS Tank, two fine mesh screens are used to create feed and discharge chambers and to separate the experimental section from these two chambers. There are two adjustable overflow pipes in the DS Tank for adjusting the water levels in the mentioned chambers and measuring the water flow. To prevent seawater intrusion, an aluminum sheet pile is used as a vertical barrier. As a result, the DS Tank has a closed water circuit with a storage tank and pump. The DS Tank and its components are depicted in **Figure 2 and Table 1**.



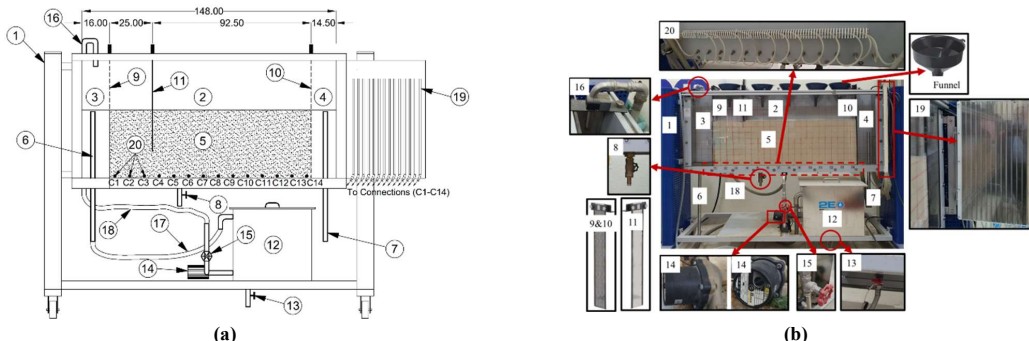

122
123

**Figure 2: DS Tank and its components: (a) Drawing of details, (b) Photo**

**Table 1: DS Tank components and descriptions**

| No. | Component Name | Description | No. | Component Name | Description |
|---|---|---|---|---|---|
| 1 | Steel frame | The DS Tank's frame | 11 | Vertical aluminum sheet pile | Vertical barrier to repel saltwater intrusion |
| 2 | Experimental section | Tank with porous media for monitoring saltwater intrusion | 12 | Storage tank | The primary source of seawater |
| 3 | Feed Chamber | Source of saltwater | 13 | Draining pipe2 | Before the next experiment, drain the saltwater from the storage tank. |
| 4 | Discharge Chamber | Source of freshwater | 14 | Pump | Pumping saltwater to the feed chamber |
| 5 | Porous media | Silica sand (0.71-1.18mm) | 15 | Pump valve | Pump flow rate adjustment |
| 6 | Outflow pipe1 | Changing the saltwater level in the feed chamber | 16 | Saltwater inflow pipe | Connecting with a pump to allow saltwater to flow from the pump to the feed chamber |
| 7 | Outflow pipe2 | Changing the level of freshwater in the discharge chamber | 17 | Hose1 | Connecting the outflow pipe 1 to the storage tank |
| 8 | Draining pipe1 | Before beginning a new experiment, drain the water from the experimental section. | 18 | Hose2 | Linking the saltwater inflow pipe to the pump |
| 9 | Vertical screen1 | Separating the feed chamber from the experimental section | 19 | 14 glass manometer tubes | Hydraulic head monitoring along the experimental section |
| 10 | Vertical screen2 | Separating the discharge chamber from the experimental section | 20 | Measuring connections | linked to the 14 glass manometer tubes |

### 2.1.2 Configuration and Experimental Set

The DS Tank and the associated materials, including saltwater, freshwater, and porous media, are pre-set for the experiments.
A horizontal and vertical scale of 5cm x 5cm is drawn on the transparent front side of the DS Tank, as shown in **Figure 3**. The
left chamber is configured as a saltwater feed chamber with a width of 16cm. The right chamber is configured as a freshwater
discharge chamber with a width of 14.5cm. Vertical screen barriers separate the experimental section of the DS Tank (length
117.5cm) from the feed and discharge chambers. The experimental section is filled to a depth of 40cm with porous media soil
(graded silica sand with grain sizes ranging from 0.71-1.18mm (see **Figure 3**). The filling process is done in layers of 5cm
each, with a falling height of 50 cm for each layer, to ensure a homogeneous hydrogeological property of the media sand. In
the filling process, funnels are used, which are distributed along the experimental section as shown in **Figure 1b**.
The seawater used in the experiments is collected from the Red Sea, and its density, as well as that of the freshwater, is
calibrated using a sensitive scale and a standard flask (see **Figure 4**). According to the calibration, the densities of saltwater



and freshwater are 0.99 and 1.022 m³/sec, respectively. In saltwater, a 0.15 g/L concentration of green food dye is used to
easily visualize the saltwater line and measure the intrusion distance inside the media sand (see **Figure 3**).

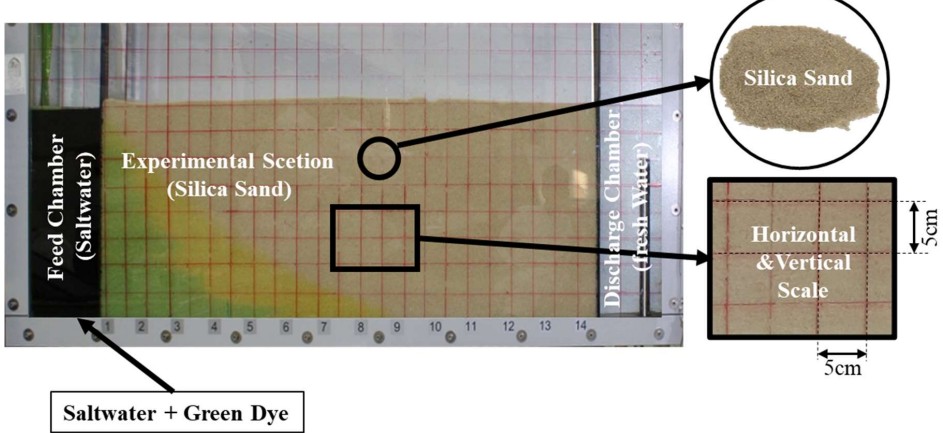

**Figure 3: DS Tank pre-set for experimental procedures**

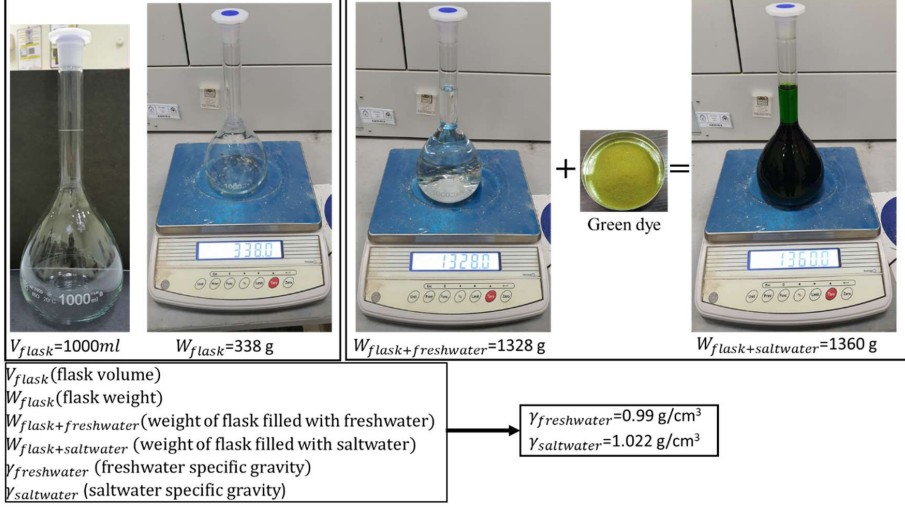

**Figure 4: Saltwater and freshwater calibration**
**2.1.3 Experimental Procedures**
The experiment procedures include the following five steps:
1-Freshwater saturation of the media sand: at the start of the experiment, the outflow pipes 1 and 2 for both the feed and
discharge chambers are set to be at the same level as the media sand surface (40 cm from the DS Tank bed). Following that,
fresh water is discharged at a constant rate into both chambers until the media sand in the experimental section is saturated.
The hydraulic heads along the experimental section are monitored by the 14 glass tube manometers until the water level reaches
the sand surface in all the manometers to verify the saturation condition.
2-Feeding the experiment with colored saltwater: in the feed chamber, an aluminum sheet pile is used to block water seepage
through the experimental section. Following that, the feed chamber's outflow pipe 1 is moved to the DS Tank bed level to
empty it of freshwater. The outflow pipe is then returned to its previous level (media sand surface level), and the storage tank
is subsequently emptied and filled with the green-dyed saltwater. When the pump is turned on and the pump valve is opened,





saltwater begins to fill the feed chamber all the way to the top of the outflow pipe1. Following that, the pump valve is manually
adjusted to maintain the saltwater level at the surface of the media sand.
3-Adjusting the water levels in the feed and discharge chambers: the first step in this process is to remove the aluminum sheet
pile from the feed chamber. Furthermore, to achieve a suitable flow through the media sand, the difference in water levels
between the feed and discharge chambers is tested several times and finally adjusted to 10 cm, resulting in a hydraulic gradient
of 0.085. To accomplish this, the outflow pipe 2 for the discharge chamber is adjusted to be 10 cm below the media sand
surface.
4-Monitoring of saltwater intrusion: in the experimental section, saltwater begins to infiltrate through the media sand and can
be observed through the transparent front side of the DS Tank. The temporal saltwater intrusion could be measured using the
horizontal and vertical scales drawn on the transparent front side. The saltwater intrusion is measured at 30-minute intervals.
Photos for each time interval are taken with a high-resolution digital camera and used to validate the observed saltwater lines
with AutoCAD software. During the experiment, the freshwater level inside the discharge chamber rises until it reaches its
maximum level by adjusting the outflow pipe2 level above the media sand surface level until it reaches a steady state.
This experimental part of the study considers two experiments:
Experiment 1 (Base Case): this is the case in which the saltwater intrusion through the media sand is studied without any
countermeasures. In this case, the procedures from steps 1 to 4 are carried out.
Experiment 2 (using a vertical barrier): through this experiment, the media sand is removed from the experimental section.
Then, the vertical aluminum sheet pile (vertical barrier) is used as a countermeasure against saltwater intrusion and placed at
the experimental section, 25cm from the feed chamber. Hereafter, media sand is refilled in the experimental section. The
penetration depth of the vertical aluminum sheet pile is set below the silica sand surface by a depth of 30cm. Then, the steps
from 1 to 4 are implemented.
**2.2 Dimension Analysis and Evaluation Ratios**
**Figure 5** illustrates the geometric properties of the experiments described in the previous section. **Table 2** summarizes and
defines several variables, parameters, and constants that influence saltwater intrusion.

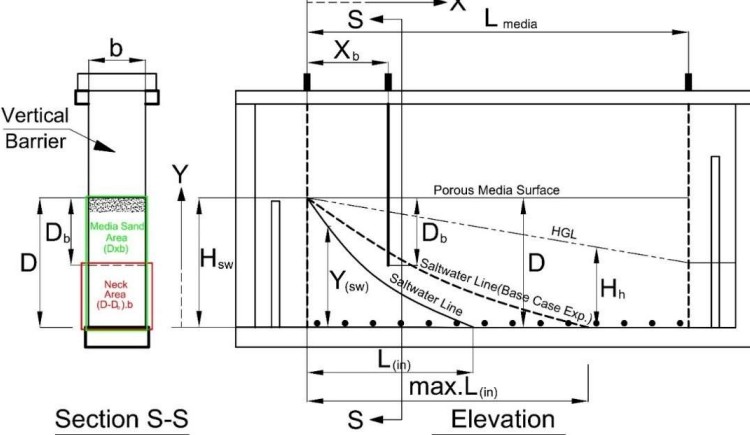

**Figure 5: Geometric characteristics of the experiments**
**Table 2: Definition of the geometric characteristics of the experiments**

| No. | Quantity | Type | | | Definition |
|-----|----------|----------|-----------|----------|------------|
| | | Constant | Parameter | Variable | |
| 1 | $H_{sw}$ | √ | | | Head of the saltwater boundary |



| 2 | $D$ | √ | | Sand media depth |
|---|---|---|---|---|
| 3 | $X_b$ | √ | | Vertical barrier location |
| 4 | $L_{media}$ | √ | | Sand media length (experimental section length) |
| 5 | $max.L_{(in)}$ | √ | | Maximum length of saltwater intrusion (attained for experiment 1 (base case)) |
| 6 | $A$ | √ | | The cross sectional area of the entire media sand |
| 7 | $a$ | | √ | Neck region (cross section area of media sand below the vertical barrier) |
| 8 | $D_b$ | | √ | Aluminum sheet pile depth |
| 9 | $X$ | | √ | Horizontal distance from the saltwater boundary measured from any embedded point in the media sand |
| 10 | $Y$ | | √ | Vertical distance measured from the experimental section bed for any embedded point in the media sand |
| 11 | $Y_{(sw)}$ | | √ | Observed saltwater intrusion depth at any X distance at a specific time (t). |
| 12 | $H_h$ | | √ | Observed hydraulic head at any X distance at a specific time (t). |
| 13 | $L_{(in)}$ | | √ | The observed length of saltwater intrusion at a specific time (t) |

Excluding the constant quantities ($H_{sw}$, $D$, $\underline{X_b}$, $L_{media}$, $max.L_{(in)}$, $A$), the function of dimensionless hydraulic and geometric parameters is represented in **Eq. 1.**

$$L_{(in)}/max.L_{(in)}= f(Y_{(sw)}/H_{sw}, D_b/H_{sw}, H_h/H_{sw}, a/A, X/L_{media}, Y/D) \tag{1}$$

**Table 3** represents the dimensionless quantities which will be considered in this study as evaluation ratios.

**Table 3: Evaluation Ratios**

| Evaluation Ratios | Definition |
|---|---|
| $L_{(in)}/max.L_{(in)}$ | Intrusion Ratio (IR): the ratio of observed intrusion length at a time (t) to maximum saltwater intrusion length (base case). |
| $Y_{(sw)}/H_{sw}$ | Salt Line Ratio (SLR): the ratio of the depth of the saltwater line at a specific distance $X$ to the depth of the saltwater boundary. |
| $D_b/H_{sw}$ | Barrier Depth Ratio (BDR): the ratio of barrier depth to the depth of the saltwater boundary. |
| $H_h/H_{sw}$ | Hydraulic Head Ratio (HHR): the ratio of the depth of the hydraulic grade line at a specific distance $X$ to the depth of the saltwater boundary.. |
| $a/A$ | Neck Area Ratio (NAR): the ratio of neck area to media sand cross section area. |
| $X/L_{media}$ | Length Ratio (LR): the ratio of horizontal distance along the experimental section to the length of media sand. |
| $Y/D$ | Depth Ratio (DR): the ratio of the vertical distance $Y$ measured from the bed of the experimental section to the total media sand depth. |

## 2.3 Numerical Model

MODFLOW-2005, in conjunction with the SWI2 package, is used in this study for numerical modeling of saltwater intrusion. SWI2 is a software package used to analyze three-dimensional groundwater flow, model saltwater intrusion, and calculate hydraulic heads. The main advantage of using the SWI2 package is that it requires fewer cells for the simulation process than variable-density groundwater flow packages like SEAWAT. The ability of SWI2 to represent each aquifer as a single layer of cells results in significant model run-time savings.

Saltwater intrusion is investigated using either a traditional vertical barrier or artificial recharge methods. Various penetration depths are simulated to control saltwater intrusion using a vertical barrier. As artificial recharge methods, surface and subsurface recharge systems are used. The above management options are evaluated separately and in combination to determine their effectiveness in addressing the saltwater intrusion problem. **Table 4** displays the numerical model cases that are investigated.



MODPATH is a post-processing package for particle tracking that computes and displays three-dimensional pathlines based on MODFLOW output. The MODPATH package is used to visualize the flow behavior of both freshwater and saltwater through the sand media by visualizing the expected transport trajectories coming from the saltwater boundary, the freshwater boundary, and the flow path from the recharge area for the cases defined in **Table 4**. When there is no vertical barrier, two water zones can be identified: zone 1 (saltwater zone) and zone 2 (freshwater zone) (see **Figure 6a**). Zones 1 and 2 are further subdivided into two zones after using a vertical barrier: zone 1a and zone 1b for saltwater and zone 2a and zone 2b for freshwater, as shown in **Figure 6b**.

**Table 4: The studied cases using numerical simulation**

| Model Cases | Description |
|---|---|
| \multicolumn{2}{c}{Category (a): using vertical barrier} | |
| Case1a | Base Case (Verification of experiment1) |
| Case2a | Vertical Barrier ($X_b$=25cm, $D_b$=35cm, $NAR$=0.125, $BDR$=0.875) |
| Case3a | Vertical Barrier (Verification of experiment2) ($X_b$=25cm, $D_b$=30cm, $NAR$=0.25, $BDR$=0.75) |
| Case4a | Vertical Barrier ($X_b$=25cm, $D_b$=25cm, $NAR$=0.375, $BDR$=0.625) |
| Case5a | Vertical Barrier ($X_b$=25cm, $D_b$=20cm, $NAR$=0.50, $BDR$=0.50) |
| Case6a | Vertical Barrier ($X_b$=25cm, $D_b$=15cm, $NAR$=0.625, $BDR$=0.375) |
| Case7a | Vertical Barrier ($X_b$=25cm, $D_b$=10cm, $NAR$=0.875, $BDR$=0.125) |
| \multicolumn{2}{c}{Category (b): using vertical barrier and surface recharge} | |
| Case1b | Base Case + Surface Recharge |
| Case2b | Vertical Barrier ($X_b$=25cm, $D_b$=35cm, $NAR$=0.125, $BDR$=0.875) + Surface Recharge |
| Case3b | Vertical Barrier ($X_b$=25cm, $D_b$=30cm, $NAR$=0.25, $BDR$=0.75) + Surface Recharge |
| Case4b | Vertical Barrier ($X_b$=25cm, $D_b$=25cm, $NAR$=0.375, $BDR$=0.625) + Surface Recharge |
| Case5b | Vertical Barrier ($X_b$=25cm, $D_b$=20cm, $NAR$=0.50, $BDR$=0.50) + Surface Recharge |
| Case6b | Vertical Barrier ($X_b$=25cm, $D_b$=15cm, $NAR$=0.625, $BDR$=0.375) + Surface Recharge |
| Case7b | Vertical Barrier ($X_b$=25cm, $D_b$=10cm, $NAR$=0.875, $BDR$=0.125) + Surface Recharge |
| \multicolumn{2}{c}{Category (c): using vertical barrier and subsurface recharge} | |
| Case1c | Base Case + Subsurface Recharge |
| Case2c | Vertical Barrier ($X_b$=25cm, $D_b$=35cm, $NAR$=0.125, $BDR$=0.875) + borewells Recharge |
| Case3c | Vertical Barrier ($X_b$=25cm, $D_b$=30cm, $NAR$=0.25, $BDR$=0.75) + borewells Recharge |
| Case4c | Vertical Barrier ($X_b$=25cm, $D_b$=25cm, $NAR$=0.375, $BDR$=0.625) + borewells Recharge |
| Case5c | Vertical Barrier ($X_b$=25cm, $D_b$=20cm, $NAR$=0.50, $BDR$=0.50) + borewells Recharge |
| Case6c | Vertical Barrier ($X_b$=25cm, $D_b$=15cm, $NAR$=0.625, $BDR$=0.375) + borewells Recharge |
| Case7c | Vertical Barrier ($X_b$=25cm, $D_b$=10cm, $NAR$=0.875, $BDR$=0.125) + borewells Recharge |





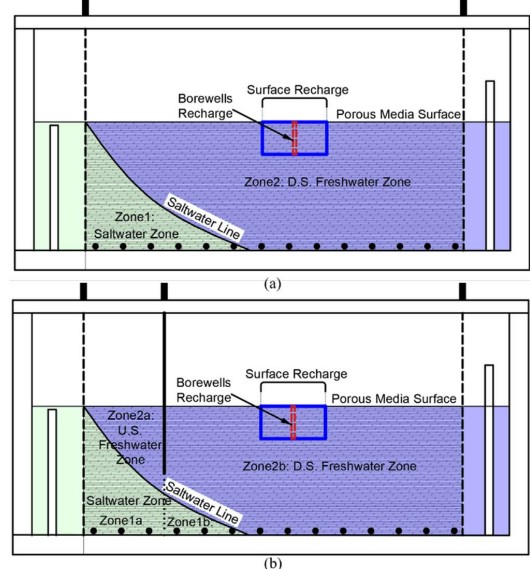

**Figure 6: Saltwater intrusion conceptual model: (a) freshwater and saltwater zones without barrier, (b) freshwater and saltwater**
**zones with barrier**
**2.3.1 Calibration and Verification Processes**
Many factors contribute to groundwater model inconsistency, including hydrogeological properties, discretization, potentially
spatial discretization, time step, and solver parameters. Using the experimental results, many trials are carried out to calibrate
the model using various hydrogeological properties, with reference to the values listed in **Table 5**. Furthermore, various
discretization schemes are tested to provide an accurate diagnosis of differences in head drawdowns and water balances. **Figure**
**7a** illustrates that 2320 cell discretization are used in this study. The transient stress period, on the other hand, is assigned to
be 120 minutes based on the experiment's monitored results, and the suitable equal interval time step is selected to be 30
minutes. The impact on the heads on the cells and the accumulated volume water balance are evaluated. Following that, a
verification procedure is implemented for:
1- Confirming the time when a steady-state condition occurs in based on the results of experiment 1.
2- Fitting the observed saltwater line in experiments 1 and 2 for the transient and steady-state conditions.
The particle tracking in the MODPATH package is simulated in the forward tracking direction using cylinder particle
placement, as shown in **Figure 7b**. in this study, the flow direction will be defined as $+^{ve}Y$, $-^{ve}Y$, $+^{ve}X$, and $-^{ve}X$, as illustrated
in **Figure 7b**.
**Table 5: Hydrogeological values of sand ((Domenico et al. 1998; Rotz 2021))**

| Soil Type | Effective Porosity | Specific Yield | Hydraulic Conductivity "K" (m/s) |
|---|---|---|---|
| Coarse Sand | from 0.18 to 0.43 | from 0.18 to 0.43 | from $9×10^{-7}$ to $6×10^{-3}$ |
| Medium Sand | from 0.16 to 0.46 | from 0.16 to 0.46 | from $9×10^{-7}$ to $5×10^{-4}$ |
| Fine Sand | from 0.01 to 0.46 | from 0.01 to 0.46 | from $2×10^{-7}$ to $2×10^{-4}$ |



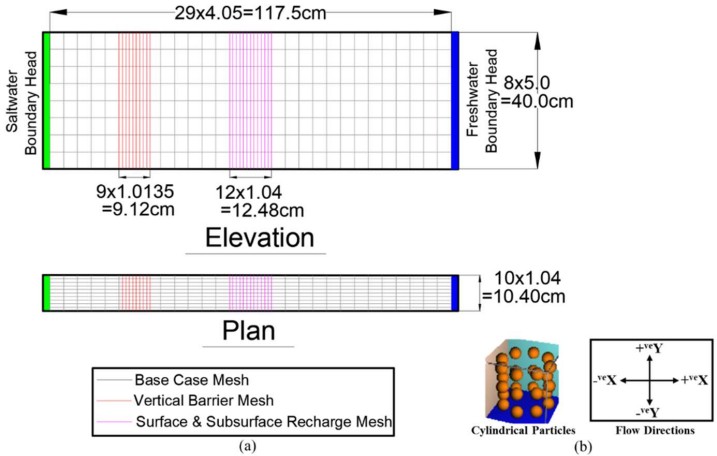

**Figure 7: Structure of the numerical model: (a) discretization and boundary conditions, (b) particle tracking and flow directions**
**2.3.2 Classification Ratios**
As a pre-set to select the best model case for repelling saltwater intrusion, four ratios are proposed to analyze the results of the
model cases included in categories (a), (b), and (c). Each ratio is calculated for each model case and then classified by its value
into best or worst. These ratios are the previously defined intrusion ratio (IR), as well as three new ratios: repulsion ratio (Rr),
wedge area ratio (WAR), and recharge ratio (RER). The last three ratios are computed using the **equations 2, 3, and 4**,
respectively with the RER ratio computed only for cases in categories (b) and (c). The criteria for classifying the best model
cases are that it has low values of IR, WAR, and RER, as well as the maximum value of Rr. on the other hand, cases with high
values of IR, WAR, and RER, as well as the lowest value of Rr, are classified as worst model case and is not recommended
for repelling saltwater intrusion. Because of the difficulty of having one of the model cases have all the best or worst values
of classification ratios to be classified as the best or worst model case (unclassified model case), it is important to use the
DMM models to use the values of these classification ratios to make this final decision.
$R_r = IR_{case1a} - IR_{casei}$ (2)
$WAR = \frac{Wedge\ Area_{case(i)}}{Wedge\ Area_{case1}}$ (3)
$RER = \frac{Recharge_{case(j)}}{Saltwater\ boundary\ Recharge_{case(j)}}$ (4)
Where case(i) is any case included at any category (a, b, and c), and case(j) is the cases included at category (b) and category
(c).
**2.4 Decision-Making Model (AHP technique)**
The AHP technique is commonly employed in decision-making systems designed to aid in decision-making and rate options
(Saaty, 1986). Actual metrics such as pricing, headcount, or subjective opinions are used as inputs into a numerical matrix in
AHP. Ratio scales and consistency indices derived from eigenvalues and eigenvectors are among the results. The AHP model
is a decision-making framework that assumes decision levels have a unidirectional hierarchical relationship (Presley, 2006).
AHP can study the interrelationships among all criteria using the hierarchical approach (Singh et al., 2007).
In this study, an AHP-based model is proposed to be employed on two levels to determine the best model case by comparing
18 model cases using numerous ratios as a selection criteria. Level (1) involves the model dealing with three categories (a, b,
and c) in order to choose the best model case among the six cases in each category. There are four criteria in case a (Rr, SLR,
HHR, and WAR), while there are five in cases b and c. (Rr, SLR, HHR, WAR, and RER). The top three model cases from
each of the three categories that emerged from level (1) can be used to create the final choice for the best model case at level




(2). Pairwise comparisons with other criteria aid in determining the relative importance of each criterion in the hierarchical
structuring of the problem. The model's first level consists of one matrix (5x5) reflecting the relative weights of the criteria
and five matrices (6x6) showing the relative weight among the alternatives in the case of each criterion. The model, on the
other hand, takes the same matrix for criteria weights and five matrices, each of which is (3x3) and expresses the relative
weight among the final three alternatives for each criterion in its second level.
The evaluations are carried out using a preference scale ranging from 1 (which represents "equally important") to 9 (which
represents "extremely important") (Saaty, 1986). The consistency ratio (CR), which is a function of the consistency index (CI)
and the relative importance (RI), is a measure of cognitive effort in the decision, and its value should not exceed 10%
(Abdelwahab et al., 2021).
**3 Results and Discussions**
**3.1 Calibration and Verification of Numerical Model**
The steady-state condition in experiment 1 (the base case) results in saltwater intrusion 90 minutes after the experiment begins.
As a time validation of the numerical model steady state simulation, **Figure 8** shows the observed and simulated saltwater
lines for various simulation times greater than 90 minutes. The figure shows that the simulated saltwater line closely matches
the observed one, with RMSE values ranging from 0.90 to 1.19 for time ranging from 90 to 120 minutes.

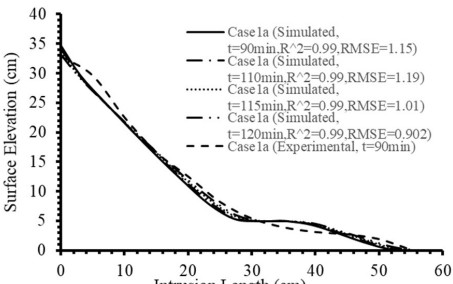

**Figure 8: Observed and simulated saltwater lines for experiment 1 (Case1a) under steady-state conditions at intervals longer than**
**90 minutes.**
For transient results, the saltwater line for experiments 1 and 2 for simulation time of 30, 60, and 90 minutes is used to verify
the corresponding results of numerical model, as shown in **Figure 9** and **Figure 10**. Both figures show that the model produces
reasonable simulated results for the saltwater lines (case3a) when compared to the observed ones. Table 6 also shows the
calibrated hydrogeological properties of the verified numerical model, including hydraulic conductivities in X, Y, and Z
directions ($k_x$, $k_y$, $k_z$), specific yield ($S_y$), specific storage ($S_s$), and effective porosity ($\eta$).

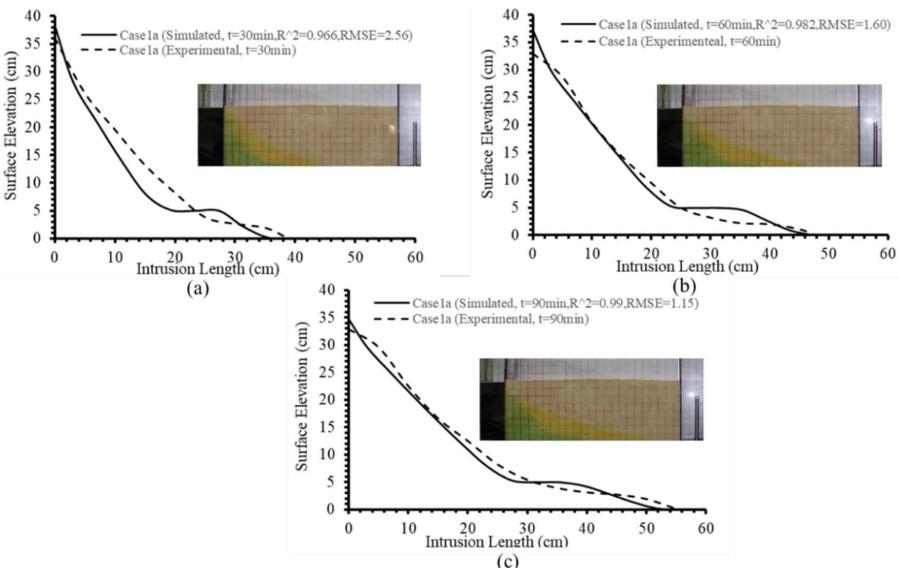

**Figure 9: Observed and simulated saltwater lines for experiment1 (Case1a) for transient state condition: (a) 30min, (b) 60min, (c)**
**90min**

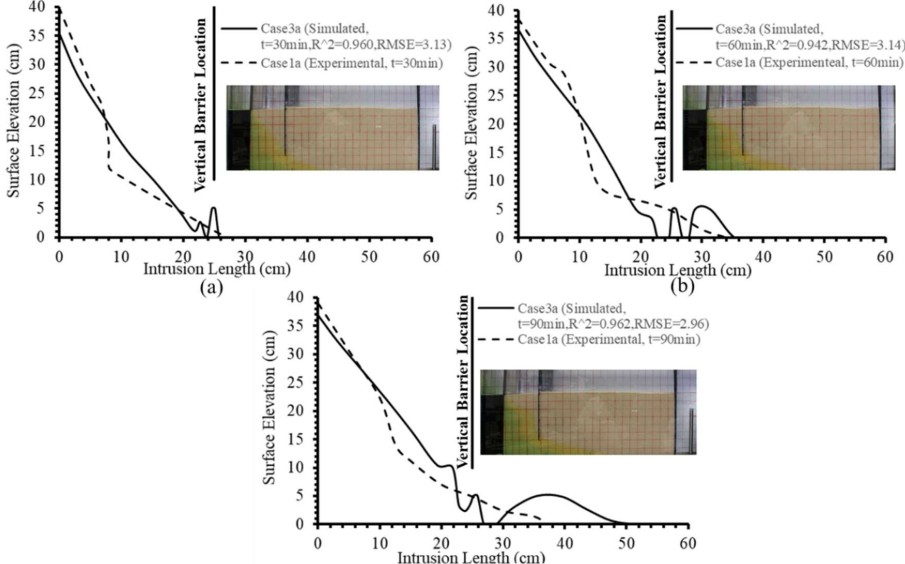

**Figure 10: Observed and simulated saltwater lines for experiment2 (Case3a) for transient state condition: (a) 30min, (b) 60min, (c)**
**90min**
**Table 6: Calibrated values of the hydrogeological properties**

| Hydrogeological Properties | $k_x$ | $k_y$ | $k_z$ | $S_y$ | $S_s$ | $\eta$ |
|---|---|---|---|---|---|---|
| Values | 0.0069 | 0.0069 | 0.03 | 0.04 | 0.0619 | 0.0428 |




### 3.2 Behavior evaluation of saltwater intrusion, flow, and hydraulic heads for categories (a), (b), and (c) model cases

#### 3.2.1 Saltwater intrusion and flow behaviors in category(a) model cases

The modeling results of saltwater intrusion and the accompanying flow behavior for the cases in category (a) will be discussed in this section, considering the evaluation ratios as summarized in **Table 3**. For case1a, **Figure 11a1** and **Figure 11a2** show that the simulated saltwater line has an IR value of 0.97 and a corresponding LR value of 0.45. **Figure a2** illustrates the flow behavior of this case, which shows that the flow in zone1 takes two directional flows: $+^{ve}$ Y and $+^{ve}$X. The $+^{ve}$Y flow causes the hydraulic heads near the saltwater boundary to be conserved at the media sand level. Furthermore, the $+^{ve}$X flow force the freshwater above the saltwater line to follow the same direction. Freshwater flow directions in zone 2 are towards $-^{ve}$X, $+^{ve}$Y, and $-^{ve}$Y. Because of the $+^{ve}$Y and $-^{ve}$Y flows, a separation line between both flows could be identified with a DR ratio in the range from 0.37 to 0.45, as shown in **Figure 11a2**. The $+^{ve}$Y flow direction conserves hydraulic head along zone2.

After adding the vertical barrier (Case2a), the IR and LR values of the saltwater line are reduced to 0.83 and 0.39, respectively, compared to those of Case1a (see **Figure 11b1** and **Figure 11b2**). Furthermore, **Figure 11b1** shows that the value of SLR along the saltwater line of zone1a decreases more than that of case1a, with a sudden drop and rise just before and after the vertical barrier. **Figure 11b2** explains this behavior by demonstrating that the high BDR (0.875) and low NAR values (0.125) impede freshwater flows from zone2a to zone2b, causing overlaying pressure on zone1a. However, when compared to case1a, the DR value of the separation line increases to be in the range from 0.40 to 0.50.

As shown in **Figure 11c1** and **Figure 11c2**, the IR and LR ratios for case3a are 0.90 and 0.42, respectively. **Figure 11c1** shows that the SLR value at zone1a slightly increases when compared to case1a. It has also been observed that the saltwater line fluctuates beneath the vertical barrier. This flow behavior is caused by the flow from zone2a to zone2b, which begins after a decrease in BDR (0.75) and an increase in NAR (0.25) compared to case2a. This explanation is supported by the flow path lines depicted in **Figure 11c2**. Because this partially flows between zones 2a and 2b, the saltwater line fluctuates and the overlaying pressure on zone 1a decreases, causing the SLR value in this zone to rise. The separation line, on the other hand, rises with DR values in the range from 0.50 to 0.68, implying that the majority of the freshwater flow is in the $-^{ve}$Y flow direction, resulting in hydraulic head reduction in zone2b.

By continuing to decrease the BDR value while increasing the NAR value, as in cases 4a, 5a, 6a, and 7a, the flow behavior for these cases follows the same pattern as in case 3a (see **Figure 11d, 11e, 11f, and 11g**, respectively). In these cases, the IR and LR ratios are gradually increased from 0.97 and 0.46, respectively, in case4a to the maximum values of 1.05 and 0.49, respectively, in case7a. Similarly, SLR values rise along zone1a until they reach the same levels as in case1a (see **Figure 11g**). Moreover, the DR value continued to increase until it reached the range from 0.75 to 0.95 in case7a resulting in a hydraulic head reduction in zone2b (see **Figure 11g2)**.

Based on the above results, it is possible to conclude that using vertical barriers with small BDR and large NAR increases SLR at zone1a as well as IR and LR values. Furthermore, the separation line with high DR values reduces the hydraulic heads along zone2b, causing an excess increase in IR and LR. Conclusively, controlling both the SLR of zone 1a and the DR of the separation line could effectively manage the saltwater intrusion represented by the IR and LR ratios. Management of saltwater intrusion will be discussed in this study by controlling the DR of the separation line as well as the hydraulic heads along zone2b using groundwater artificial recharge associated with the use of a vertical barrier.



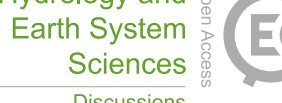


(a1)    (a2)


(b1)    (b2)


(c1)    (c2)


(d1)    (d2)


(e1)    (e2)



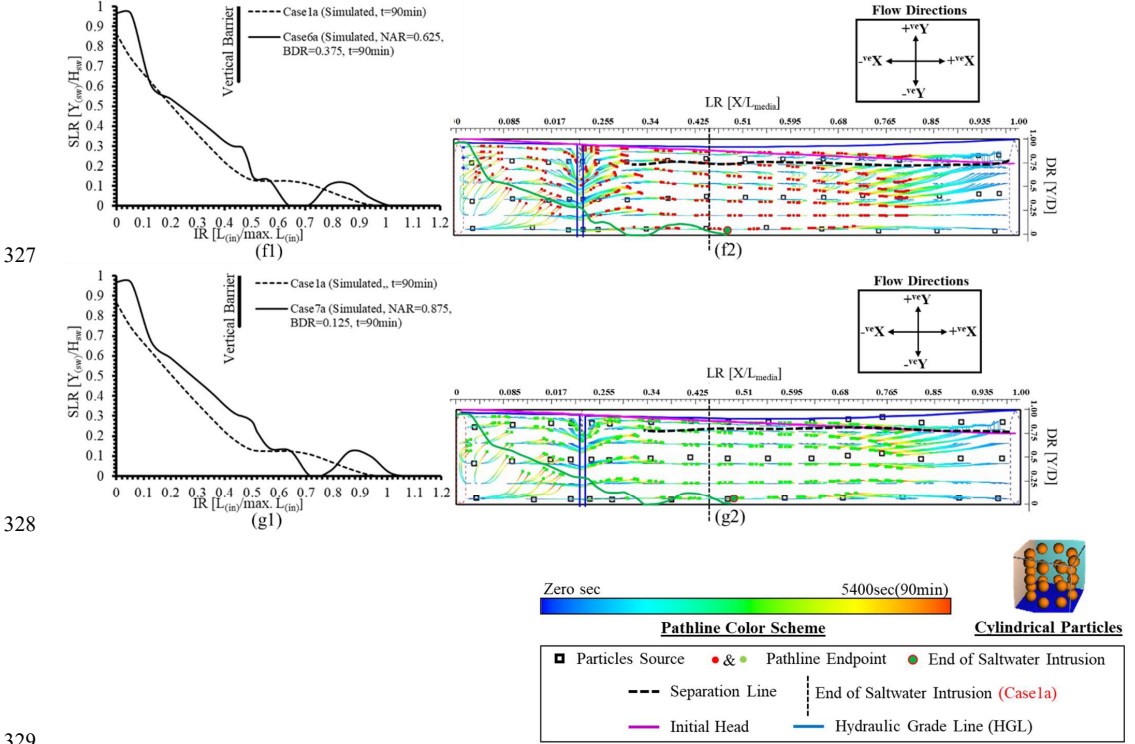



Figure 11: Simulated saltwater lines and groundwater flow behavior of the category (a) model cases: (a) case1a, (b) case2a, (c) case 3a, (d) case4a, (e) case5a, (f) case6a, (g) case7a

### 3.2.2 Hydraulic head variations in category(a) model cases

The hydraulic head variations are analyzed for the category(a) modeling cases, as shown in **Figure 12**. This figure illustrates the relationship between HHR and LR ratios with displaying the minimum HHR values and their locations along the aquifer.

**Figure 12** shows that the hydraulic head of case7a has the lowest HHR value of 0.91 compared with the other cases (cases 1a-6a) located at a LR value of 0.55 (see **Figure 12g**). On the other hand, Case1a, has the highest value of the minimum HHR (0.98), and a location has a LR of 0.44, as shown in **Figure 12a**.




**Figure 12: Values and locations of the minimum HHR of the category (a) model cases: (a) case1a, (b) case2a, (c) case 3a, (d) case4a,**
**(e) case5a, (f) case6a, (g) case7a**
The above results can be summarized as illustrated in **Figure 13**, which depicts the effect of BDR on the location (LR) and
value of the minimum HHR and the IR ratios. The minimum hydraulic head is located at zone2b for all study cases (case1a-
case7a), with LR values ranging from 0.45 to 0.55 and corresponding minimum HHR values ranging from 0.91 to 0.98. On
the other hand, the maximum IR, occurs for both cases 6a and 7a with a value of 1.05 when using a BDR in the value range
from 0.25 to 0.38. Given these findings, increasing the hydraulic head represented by HHR could effectively repel saltwater
intrusion when combined with the vertical barrier countermeasure. For this purpose, using groundwater artificial recharge,
whether by surface or subsurface recharge, at the location of the minimum HHR value (LR in the range from 0.45 to 0.55),
combined with the use of a vertical barrier, could be used to repel saltwater intrusion, as will be discussed in the following
sections of this study.



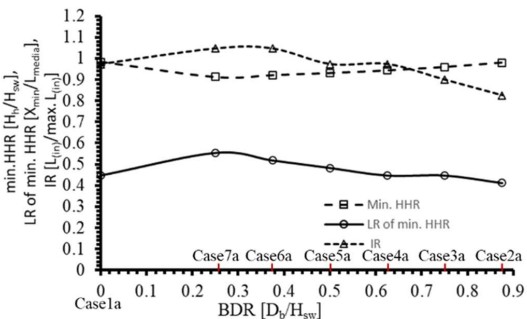

**Figure 13: Effect of BDR on the IR and minimum HHR values and locations**
**3.2.3 Saltwater intrusion and flow behaviors in categories (b) and (c) model cases**
Groundwater artificial recharge is used to repel saltwater intrusion in zone2b along the LR range (from 0.45 to 0.55), which
has a minimum value of HHR for preserving its value at the unity value. Surface and subsurface recharge are numerically
discussed, either separately or in conjunction with the vertical barrier, as shown in **Table 4** for category (b) and category (c)
study cases. The recharge is applied along the whole range of LR values from 0.45 to 0.55 for surface recharge. In contrast,
for subsurface recharge, the recharge is applied as a line of wells at the midpoint of the same LR range at a value of 0.5. The
results of category (b) and category (c) study cases will be compared with the base case results (case1a) and the corresponding
cases of category (a) in the following discussions.
**Figure 14a1** and **Figure 14a2** show that the saltwater line in case1b has IR and LR values of 1.00 and 0.47, respectively,
which are slightly higher than those in case1a (0.97 and 0.46, respectively). When comparing case 1c to case 1a, the IR and
LR values increased to 1.05 and 0.49, respectively. For both model cases 1b and 1c, it is observed that the value of SLR
increased more in both cases than in case 1a. This salt line behavior could be explained by the fact that, in case1b, the surface
recharge works as a hydraulic barrier that prevents saltwater from flow in the $^{+ve}X$ direction as well as forces it to flow
intensively in the $^{+ve}Y$ direction. This behavior causes an increase in the SLR (increase in saltwater head), which leads to an
increase in the IR and LR values, and the majority of recharged freshwater is forced to take a $^{+ve}X$ direction (see **Figure 14a2**).
The flow behavior in case1c is the same as in case1b (see **Figure 14a3**); however, the countermeasure effect of subsurface
recharge, which is a line of wells, is less than that of surface recharge, which is a water mass. This could explain the higher IR
and LR values in case1c compared to case1b, with the same SLR value.
The combined effect of the vertical barrier and surface recharge repels saltwater intrusion significantly in case2b, with IR and
LR values of 0.75 and 0.35, respectively, compared to corresponding values of 0.83 and 0.39 in case2a (see **Figure 14b1** and
**Figure 14b2**). In contrast to case 2a, although the value of SLR increases in zones 1a and 1b, the $^{-ve}X$ flow direction of surface
recharge towards the neck area under the vertical barrier forces the saltwater line to repulse (**Figure 14b2**). Because of the
lower effect of well recharge than that of surface recharge, the IR and LR ratios are higher in case 2c than in case 2b, with
values of 0.82 and 0.38, respectively. Conclusively, the IR, LR, and SLR values for Case 2b are the lowest compared with
those of Case 2c and the previous cases (Case 1b and Case 1c) confirming the efficient combination of the vertical barrier and
surface recharge at the location of the minimum HHR (LR in the range from 0.45 to 0.55).
**Figure 14c1** and **Figure 14c2** show that the IR and LR values for case3b are significantly lower, at 0.68 and 0.32, respectively,
when compared to the corresponding values for case2b and case3a. On the contrary, the SLR value increases in comparison
with the same cases, however, it drops dramatically just behind the vertical barrier of zone 1a, as shown in **Figure 14c1**.
According to one interpretation of this behavior, because of the high NAR value (0.25) compared to cases 2b and 3a, freshwater
flows intensively from zone 2a to zone 2b, causing a decrease in SLR value in the adjacent area to the vertical barrier as well
as fluctuations beneath the barrier (see **Figure 14c2**). The IR and LR ratios for case3c are 0.82 and 0.38, respectively, which





is quite large when compared to case3b. Furthermore, due to the weak effect of well recharge, the SLR value of zone1b is
greater than that of case3b (see **Figure 14c1** and **Figure 14c3**).
By continuing to decrease BDR while increasing NAR, IR and LR values increase significantly compared to the previous case
(case3b) to have the same values of 0.81 and 0.38 for cases 4b, 5b, 6b, and 7b. Similarly, IR and LR values increase
significantly when compared to case 3c, reaching 0.85 and 0.4, respectively, for cases 4c, 5c, and 6c, and 0.75 for case 7c
(**Figure 14d1, e1, f1, and g1**). This behavior is due to the freshwater's $+^{ve}X$ flow direction from zone 2a to zone 2b, which
reduces the effect of surface and well recharge, as shown in **Figure 14d2, d3, e2, e3, f2, f3, g2, and g3**. In comparison to the
corresponding IR and LR values for cases 4a, 5a, 6a, and 7a, which are in the range from 0.97 to 1.04 and from 0.46 to 0.49,
respectively, the corresponding IR and LR values for category(b) and category(c) are smaller, as shown in **Figure 14d1, e1,**
**f1, and g1**. The hydraulic heads along the experiment section are unchanged in all cases of categories (b) and (c), and the DR
ratio of the separation line is nearly the same with a value range from 0.75 to 0.90.
Based on the findings, it is possible to conclude that artificial aquifer recharging along the LR values from 0.45 to 0.55, which
has a minimum value of HHR ratio to conserve its value, as well as the unity accompanied by using a vertical barrier, has a
significant effect on saltwater line repulsing. Furthermore, because of its body mass, surface recharge is more efficient than
well recharge.

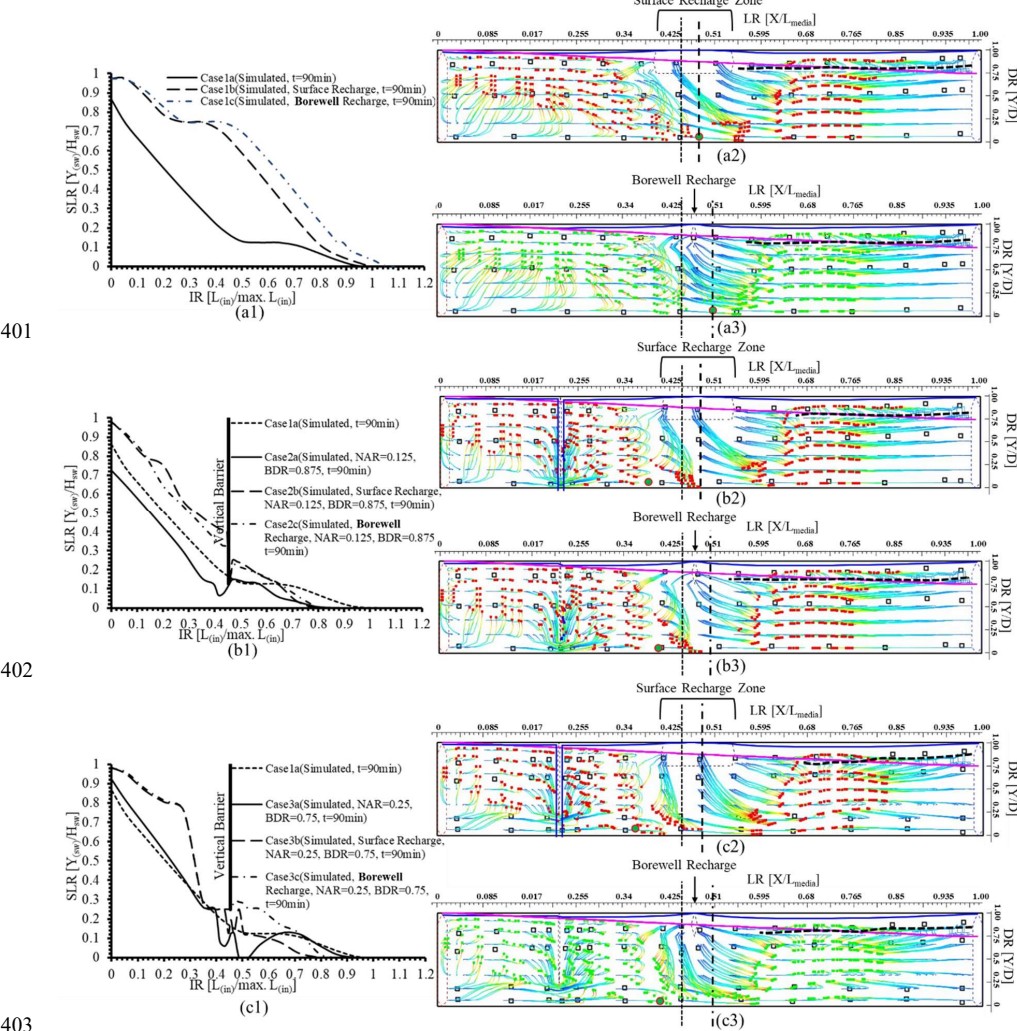

















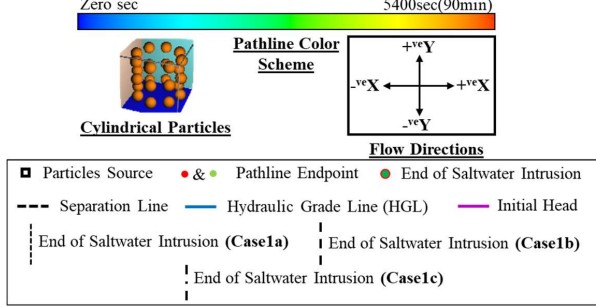

**Figure 14: Simulated saltwater lines and groundwater flow behavior of the category (b) and (c) model cases: (a) case1b &1c, (b)**
**case2b&2c, (c) case 3b&3c, (d) case4b&4c, (e) case5b&5c, (f) case6b&6c, (g) case7b&7c**
**3.2.4 Hydraulic head variations in categories (b) and (c) model cases**
**Figure 15a** to **Figure 15g** depict the hydraulic heads along the aquifer as represented by the HHR ratio for cases in categories
(b) and (c) compared to category (a). The hydraulic heads for all cases have been conserved along the LR ratio from 0.45 to
0.55, which has the minimum value of HHR to have the unity value, and the losses through the vertical barrier are greatly
reduced when compared to those of category (a).

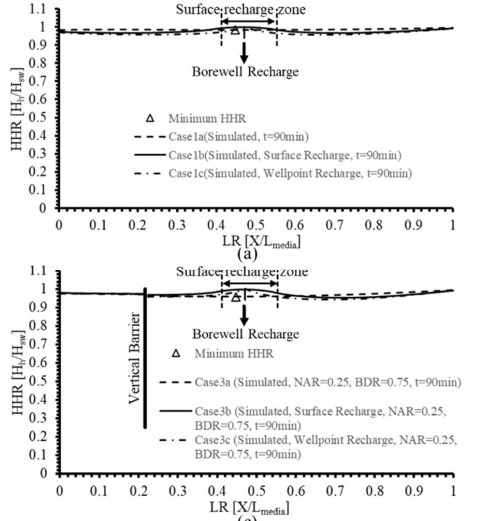

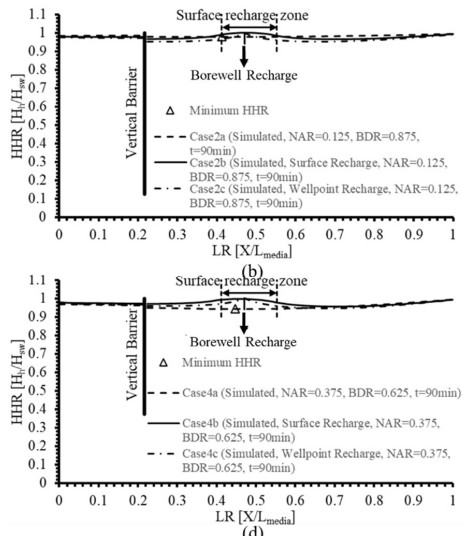




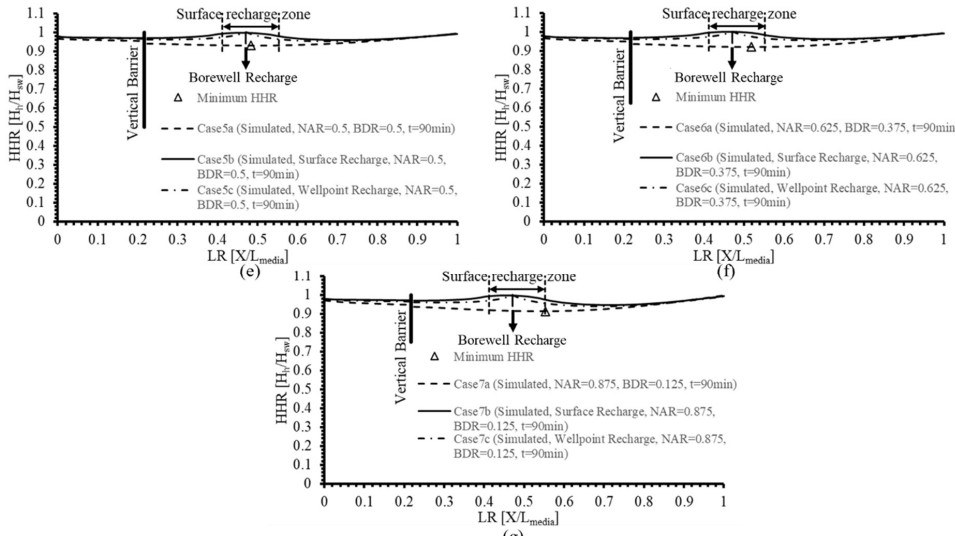

**Figure 15: Hydraulic head variation along the aquifer for categories (b) and (c) compared with those of category (a) model cases:**
**(a) case1a&1b&1c, (b) case2b&2b&2c, (c) case 3b&3b&3c, (d) case4b&4b&4c, (e) case5b&5b&5c, (f) case6b&6b&6c, (g)**
**case7b&7b&7c**
**3.3 Classification of model cases**
The classification ratios described in Section "2.3.2 Classification Ratios" are summarized and classified in **Figure 16 and**
**Table 7**. **Figure 16a** presents the IR and Rr values for each model case, whereas **Figure 16b** depicts the WAR and RER
values. **Figure 16** and **Table 7** show that case3b has the best IR and Rr values of 0.68 and 0.29, respectively. Case2a, on the
other hand, has the highest WAR value of 0.76. Furthermore, case7c has the highest RER value of 1.91. **Figure 16** and **Table**
**7** show that case1c has the worst IR, Rr, and WAR values of 1.05, -0.07, and 2.18, respectively. Furthermore, case6b has the
lowest RER value of 3.62. The remaining model cases are categorised as unclassified model cases. Based on the findings, it is
unable to determine which model case is the most successful scenario to implement as a saltwater intrusion countermeasure.
As a result, the DMM model is critical for determining the most effective model case.

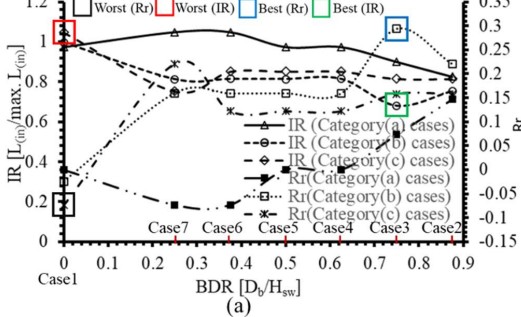






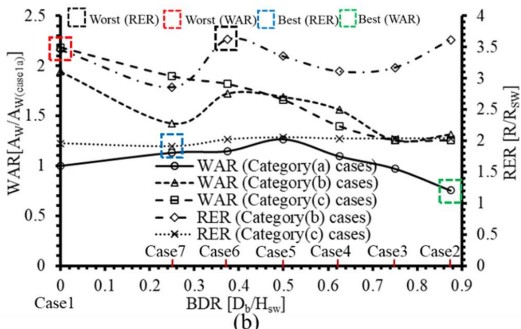

**433**
**434** **Figure 16: Classifications of model cases included in categories (a), (b), and (c): (a) IR and $R_r$ values, (b) WAR and RER values**

**435** **Table 7: Model cases classification according to values of classification ratios**

| Classification Ratio | Best | Worst |
|---|---|---|
| IR | Case3b (0.68) | Case1c (1.05) |
| $R_r$ | Case3b (0.29) | Case1c (-0.07) |
| WAR | Case2a (0.76) | Case1c (2.18) |
| RER | Case7c (1.91) | Case6b (3.62) |

**436**

**437** **3.4 Selecting the Most Effective Model Case (AHP application results)**

**438** As previously stated, the AHP model is applied to the numerical model results at two different levels of selection (levels (1)

**439** and (2)). Model cases are referred to as alternatives at this stage, and selected ratios among the evaluation and classification

**440** ratios are referred to as criteria. Level (1) needs for determining the best alternative (model case) in each category. Furthermore,

**441** level (2) is for deciding the best model case. For category (a) model cases (alternatives), the criteria values used include SLR,

**442** HHR, $R_r$, and WAR, and the RER is added over these ratios for categories (b) and (c).

**443** **3.4.1 Level (1) Results**

**444** For the alternatives in each category, the model is applied at level (1) using the weights chosen among criteria as shown in

**445** **Figure 17**. According to **Figure 17a**, HHR has the largest weight for category (a) alternatives, followed by $R_r$. Also, WAR

**446** has the lowest weight. Similarly, for category (b) alternatives (see **Figure 17b**), the same rating is observed for HHR (the

**447** highest weight), followed by $R_r$, while WAR has the lowest weight.

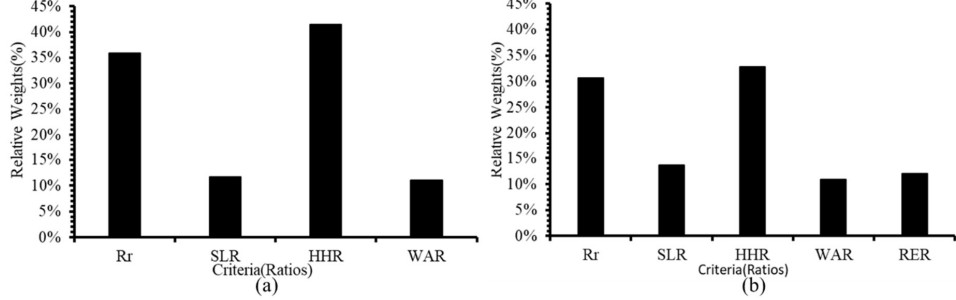

**448**
**449** **Figure 17: Level (1) criteria relative weights for different categories: (a) category (a), (b) category (b), and (c)**

**450** **Figure 18** illustrates the results of the relative weights in the three categories for each alternative. It is evident that case 2a

**451** ranks first, with a relative weight of 38.72%, followed by case 3a, while case 6a ranks last in this category (see **Figure 18a**).

**452** Case 3b is the best alternative in category (b), followed by Case 4b, which has a weight difference of 3.5% with Case 3b. and

**453** Case 6b is the worst alternative in this category (see **Figure 18b**). Case 7c is the recommended alternative in category (c), with





a weight of 25.83% ahead of the rest of the alternatives, followed by Case 4c, while Case 6c placed in last place in this category
(see **Figure 18c**).

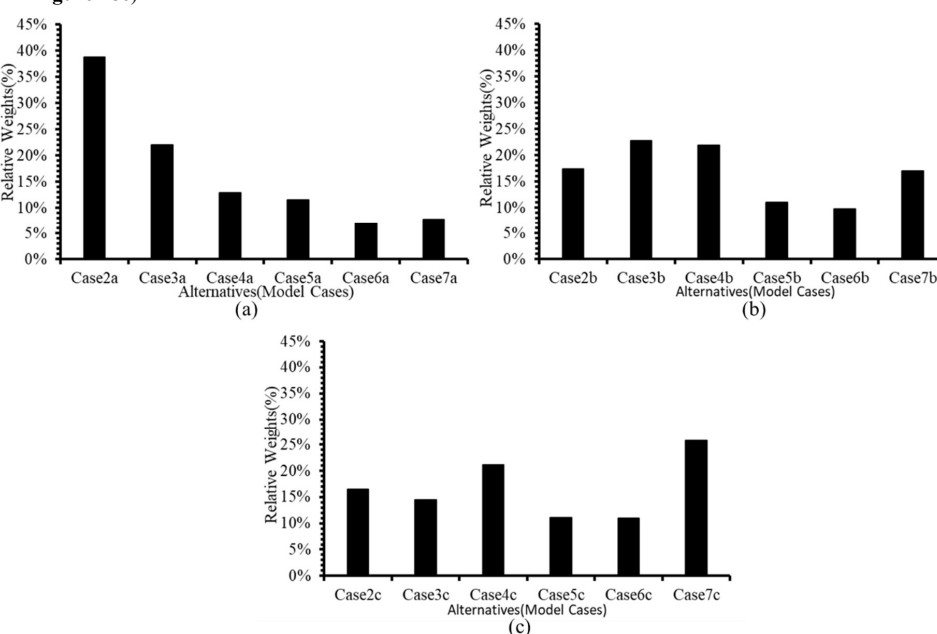

**Figure 18: Level (1) relative weights among alternatives: (a) category (a), (b) category (b), (c) category (c)**
**3.4.2 Level (2) Results**
The model's level (1) findings are summarized in the three best case model alternatives (cases 2a, 3b, and 7c), as shown in
**Figure 18**. **Figure 19** summarizes the relative weights for each criterion in relation to the three alternatives. In case 2a, SLR
is the most effective criterion, followed by WAR, while RER has a negligible effect (**Figure 19a**). on the other hand, Rris the
most essential parameter influencing case 3b, followed by HHR (**Figure 19b**). Case 7c is clearly influenced primarily by RER
(**Figure 19c**).




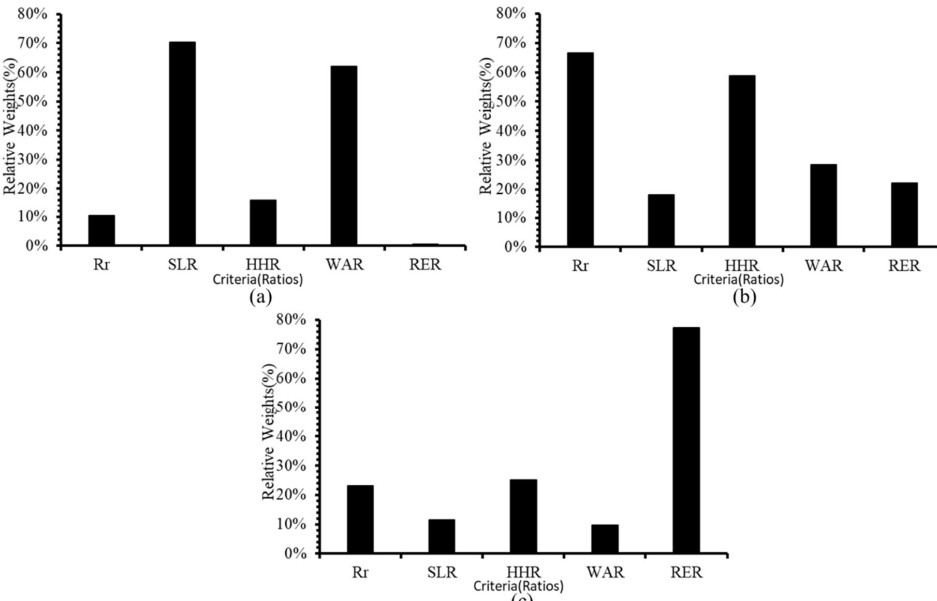

**Figure 19: Level(2) relative weights for each criterion of the alternatives for final decision: (a) case2a, (b) case3b, (c) case7c**
As a result of the preceding findings, **Figure 20** illustrates the weight values of the alternatives as a final decision, which
clearly supports Case 3b by a percentage of 47.93% over a percentage of 27.30% for Case 7c and 24.85% for Case 2a. Based
on the findings, it could be conclude that the components of case3b (combining the vertical barrier with surface recharge along
the LR ratio from 0.45 to 0.55) could be classified as best model case for use as a saltwater repellent countermeasure.
Furthermore, the vertical barrier has a greater effect when combined with surface recharge than when combined with well
recharge. On the other hand, surface recharge necessitates a high recharge rate (about 1.25 times the borewell recharge).

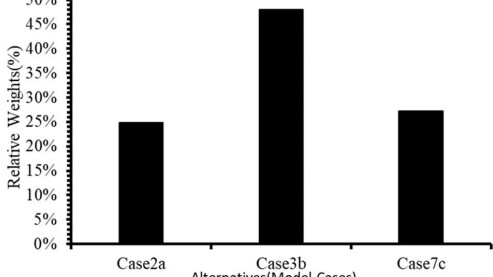

**Figure 20: Level (2) relative weights for the three alternatives for final decision**
**Conclusion**
Seawater intrusion is a common environmental issue that degrades the quality of fresh groundwater in the coastal aquifer.
Because of the hydraulic connection between the coastal aquifer and the sea, using conventional physical vertical barriers
could reform the groundwater's hydraulic gradient, disrupt the hydrodynamic balance between the two fluids, affect the
potentiometric surface of the coastal aquifer, and increase saltwater intrusion. In this study, saltwater intrusion is managed by
controlling hydraulic heads along the coastal aquifer using surface or subsurface recharges in conjunction with the traditional
vertical barrier countermeasure. A physical model is created to investigate the saltwater line behavior with a vertical barrier
(experiment1) and without a vertical barrier (experiment2). The experimental results are used to validate a MODFLOW created
numerical model. Following that, three categories of model cases ((a), (b), and (c)), each with seven numerically proposed





model cases, are numerically proposed for: analyzing the saltwater-freshwater interaction through porous media; selecting the
best location of the recharge; determining the best depth of the vertical barrier; and selecting the components of the efficient
countermeasure system, including the vertical barrier, surface recharge, and subsurface recharge. The dimension analysis
technique is used to generate evaluation ratios in order to analyze and characterize the numerical model cases' saltwater line
and hydraulic head variations. According to category (a) simulation results, the minimum hydraulic head occurs through length
ratio (LR) values ranging from 0.45 to 0.55 with corresponding values of hydraulic head ratio (HHR) ranging from 0.91 to
0.98. On the other hand, surface and subsurface recharge, are implemented through categories (b) and (c) to investigate
saltwater repulsion by maintaining the HHR value of unity within the concluded LR range. As a preset for finding the best
model case, classification ratios are proposed to classify the model cases included in the three mentioned categories as the best
or worst model case. Using the calculated classification ratio values, an analytic hierarchy process (AHP) decision-making
model (DMM) is used to select the best model case that is recommended for saltwater repulsion using two selection levels.
The first selection level concluded that the HHR has the highest relative weight in all categories, while the WAR has the
lowest. Cases 2a, 3b, and 7c are rated as the best model cases in categories (a), (b), and (c), respectively, and are most affected
by SLR, Rr, and RER, respectively. In the second selection level, the final decision is made that case 3b is the overall best
model case, which has a reasonable WAR of 1.25 and a minimum IR and maximum Rr of 0.68 and 0.29, respectively.
Moreover, the findings indicate that countermeasure systems (combining the vertical barrier with surface recharge) are the
best choice to be used in this case.

**Data availability:** The authors support the data availability of this research, which includes experimental measurements,
numerical model output files, numerical model post-processed results, and AHP output results. The data source can be found
at https://doi.org/10.4211/hs.8c31e2e9f8ab459ab99c61ccc110ab08 (Mahmod, 2023).
**Author contribution**: "Wael Elham Mahmod designed the experiments and carried them out. He also built the numerical
model using ModelMuse and performed the simulations and analysed the results. With contributions from all co-authors, Wael
Elham Mahmod prepared the manuscript. Usama Hamed Issa developed the AHP model and performed the decision-making
simulations.
**Financial Support:** This research work was funded by Institutional Fund Projects under Grant No. IFPIP-1252-137-1443
provided by the Ministry of Education and King Abdulaziz University, DSR, Jeddah, Saudi Arabia.
**Acknowledgements:** The authors gratefully acknowledge the technical and financial support provided by the Ministry of
Education and King Abdulaziz University, DSR, Jeddah, Saudi Arabia.
**Competing interests**: The contact author has declared that none of the authors has any competing interests.

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
