# Peer review of "Developing Functional Recharge Systems to Repel Saltwater"

_Hydrology and Earth System Sciences, 2023_

## Referee Comment (RC1)

**Review of**

**Developing Functional Recharge Systems to Repel Saltwater Intrusion via Integrating Physical, Numerical, and Decision-Making Models for Coastal Aquifer Sustainability**
By

**1. Introduction and overall assessment**

The authors assess the improvement caused by adding artificial recharge to an impervious barrier as measures to control seawater intrusion (SWI). To this end, they perform sandbox experiments and numerical methods on a highly idealized case. They use this model to compute a number of ratios that are used to feed an "analytic hierarchy process". They conclude that the combination of physical barrier with artificial recharge is the best option to control SWI.

Overall, the topic is of interest to HESS readers and the conclusion is valid, although somewhat trivial. Since they are not considering construction or operation costs, the more barriers you put (physical and or hydraulic), the better you will control SWI. Furthermore, it is clear that the authors have put a lot of work into this paper. Unfortunately, I believe the paper cannot be published in the present form for several reasons:

1) It is unnecessarily complex and, worse, incomplete. A large number of indicators are defined without a clear reason (no formal dimensional analysis). Yet, the most important parameters (recharge rate) are not defined.
2) The paper contains numerous conceptual errors (not too severe, but unacceptable).
3) It is very poorly written. Worse, it is very poorly organized. I do not just mean the language needs to be improved, but also the logical sequence.

I discuss issues 1) and 2) below and I make a number of writing recommendations in the last section. But the overall recommendation would be to greatly simplify your paper and remove everything that is not related to the main objectives.

**2. Conceptual errors:**

**Introduction**

Line 58: It is not true that "Physical and numerical models … reduce the high cost of hydrogeological and environmental investigations". There is no alternative to scu investigations. If you design a SWI control system without a good understanding of you system, you will waste you money (what is the depth?, how do you know you are intercepting the whole SW flux?, how much you need to recharge?

**Methods:**

I am not sure what you mean by "dimension analysis", but none of the dimensional analyses" I know consist of computing model output ratios. Please, do not anticipate results before describing what you do ("The results of the category (a) model cases reveal the location of the minimal hydraulic heads, which are expected to be the locations of the indicated artificial recharge systems" (and we still do not know what model category (a) is).

**Sand tank:**

Note that the selected ratios are arbitrary and do not result from a proper "dimensional analysis". That is, appropriate rations would result from writing the problem in dimensionless form, so that they would represent the minimum set of variables for definition of the problem. The current definitions are (1) arbitrary (e.g., IR is defined with respect to a base case that has not been defined yet, perhaps it would be better to define it with respect to the case without any remediation), (2) redundant (NAR and NDR are complementary, except that, to make things worse, BDR is defined in terms of cross sectional area), (3) not really descriptive variables (e.g., the SLR is not a real number, but a function). This is severe, as it invalidates the final results. Worse, the reader is left with the impression that the ratios are improvised. After devoting some length to describing these ratios in section 2.2, the authors introduce new ratios in section 2.3. Worse, the numbering of the new ratios is inadequate.

Recharge through wells is done when you have an aquitard. Furthermore, wells generate a radial flow around the well. The setting (both the sandbox and the model) are essentially 2D, so that including wells is not appropriate.

Further comments:

Line 132: "The filling process is done in layers of 5cm each, with a falling height of 50 cm for each layer, to ensure a homogeneous hydrogeological property of the media sand". Actually this procedure may lead to stratification (it depends on the filling rate) with the coarsest material at depth. While this may be adequate (it is similar to what occurs in nature) it does not ensure "a homogeneous ... sand".

Densities of 0.99 and 1.022 kg/L (not m3/s) are low and high for FW and SW, respectively. Adding 0.15 g/L concentration of green food dye will increase the density of SW.

Line 149: You are not verifying saturation, hydraulic heads will equilibrate even if air bubbles are trapped.

Figure 4 can be dropped. It is not necessary.

In Figure 5, name only variables that you are going to vary (b is fixed, isn't it?). Also you may add the location of recharge wells here, so that you can eliminate Figure 6.

Tables 1-3 can be blended into 1 (if at all). Table 3 is particularly unfriendly s it forces the reader to check that all cases are identical. It took me a while to realize that you were just varying the barrier depth. In fact, you may just say that you did several runs with varying barrier depth and make them a function of barrier depth.

Please, eliminate Table 5 (textbook and irrelevant for your work)

**Numerical Model**
The model is not properly described. Beyond formalities, this is severe because the results suggest that boundary conditions in the model /never described) are different to those in the sand tank.

**AHP technique**

The description is too sketchy (I had to read independently to understand it). While, in my understanding, the AHP technique is not appropriate here. It is generally used for multiple criteria that are hard too subjective for quantitative comparison, so that the weights assigned to each criteria are derived from pair-wise comparisons by "experts".

**Results**

Model calibration is unclear. For one thing, the authors report a RMSE without having described what errors are being minimized. The parameters results are unrealistic (Table 6 contains no units, I assume that k is m/s, but a porosity of 0.04, a specific storage of 0.0619 1/m?, and a $S\_Y$ of 0.04 are clearly unrealistic). I do not think that these errors affect results severely, but it conveys a poor image of the model. Probably more severe are the apparent instabilities or the fact that boundary conditions look different. IT is also unfair to the reader to find here, for the first time, that model construction is done under transient conditions, which leaves the reader wondering how are the indices described in Section 2 computed.

Worse, the flat region of the salt interface (between 30 and 40 cm inland) in the experiment suggests that permeability is higher in this region (probably a consequence of the handling of the barrier). This flat region was reproduced in the numerical model, but nothing is said about heterogeneity.

The graphs do not appear to make much sense. It is not clear to me what hydraulic grade line is. But if I interpret it to mean head, the paths are inconsistent. You probably do not need the paths.

Results of each case are hard to read because of the abuse and redundancy of indicators and inconsistencies. For example, IR had been defined the ratio of observed intrusion length at a time (t) to maximum saltwater intrusion length (base case). Yet, for the base case IR is found to be 0.97.

I could not follow the last part (new items are introduced and I must confessed that I was exhausted of going back and forth to recall the meanings of all the abbreviations).

**Editorial comments and writing**

The paper is very poorly written. I am referring not only to the traditional "look for a native speaking person" (being a non-native myself, I hate when I am told it), but also to the logic. The paper is complex and long, and the writing does not help. Try to simplify it. I understand that you are under pressure to publish quickly, but, please, facilitate the lives of readers by providing an easy to read paper (in the end, it will favor citations). Numerous statements call for a more refined argumentation. I list a few below:

In the abstract: "Three countermeasure combinations, including vertical barrier, surface, and subsurface recharges, are numerically investigated using three model case categories. Category (a) model cases investigate the hydraulic head's variation along the aquifer to determine the best recharge location. Under categories (b) and (c), the effects of surface and subsurface recharges are studied separately or in conjunction with a vertical barrier". Perhaps it is sufficient to say "The numerical model is used to investigate the SWI control efficiency of vertical barrier, and optimally located surface and subsurface artificial recharge".

Also, in the abstract, try to minimize the use of acronyms (OK in the body of the paper) and too many numbers (they tend to hide, rather that highlight). For example, it is very hard to read: "An analytic hierarchy process (AHP) as DMM is built using the classification ratios of hydraulic head (HHR), salt line (SLR), intrusion (IR), repulsion (Rr), wedge area (WAR), and recharge (RER) as selection criteria to select the overall best model case. The optimal recharging location, according to the results, is in the length ratio (LR) range from 0.45 to 0.55. Furthermore, the DMM supports case3b (vertical barrier + surface recharge) as the best model case to use, with a support percentage of 47.93%, implying that this case has a good numerical model classification with a minimum IR of 67.9%, a maximum Rr of 29.4%, and an acceptable WAR of 1.25". Instead, you may just say: "An analytic hierarchy process is built to compare SWI control strategies on the basis of head, salt line, intrusion, repulsion, wedge area and recharge. We find that best results are obtained by combining a vertical barrier with surface recharge at a distance from the coast comparable to the thickness (here you have a problem, your LR is a model dependent variable, you must relate it to generic aquifer variables".

The introduction is very poorly structured... You start by saying that SWI is a relevant problem, continue with methods to control SWI (but do not mention artificial recharge, which many, including me consider the best control method, as its efficiency is greater than 1, see Abarca et al., WRR, 2006). Then you introduce artificial recharge to conclude that "Although many studies investigate saltwater intrusion in coastal aquifers, only a limited number study the control methods of saltwater intrusion"! This is inconsistent (and false!). I believe that the logical sequence of the introduction needs to be revised.

Line 46: "Artificial recharge techniques, such as surface and subsurface recharge systems, are critical for establishing hydraulic barriers and mitigating the effects of saltwater intrusion". They are not critical... Instead "Artificial recharge techniques can be used for establishing hydraulic barriers and mitigating saltwater intrusion, while recovering SGD".

"These techniques have several advantages compared to traditional methods, including low cost..." what traditional techniques do you refer to?

"Although many studies investigate saltwater intrusion in coastal aquifers, only a limited number study the control methods of saltwater intrusion". Indeed, you should cite some of them

"Although physical and numerical models are effective economic tools for selecting the best solutions for repelling saltwater intrusion, deficiencies in the acquisition of appropriate evidence to support the final decision are discovered". What do you mean by discovered... Perhaps you should indicate some of these deficiencies.

 The objective statement should be short and to the point. Instead, you list seven goals, which are really methodological steps.

Look for appropriate references (this may also help you to simplify the writing by leading the reader to other papers for details). For example, "MODFLOW-2005, in conjunction with the SWI2 package, is used in this study for numerical modeling of saltwater intrusion. SWI2 is a software package used to analyze three-dimensional groundwater flow, model saltwater intrusion, and calculate hydraulic heads. The main advantage of using the SWI2 package is that it requires fewer

cells for the simulation process than variable-density groundwater flow packages like SEAWAT. The ability of SWI2 to represent each aquifer as a single layer of cells results in significant model run-time savings". I am not sure this choice is appropriate, but please provide references for all the codes.

Also, provide a reference for "HM 169 GUNT HAMBURG"

There are numerous terms that you must revise (you do not "repel seawater intrusion", you control it, or minimize it)

---

## Author Comment (AC1)

**Developing Functional Recharge Systems to Control Saltwater Intrusion via Integrating Physical, Numerical, and Decision-Making Models for Coastal Aquifer Sustainability**

Yehia Miky[1], Usama Hamed Issa[2,3], Wael Elham Mahmod[3,4]

[revised manuscript text omitted]

Physical and numerical models have not only proven to be more effective tools for selecting the optimum solutions for controlling saltwater intrusion but can also be used to reduce the need for expensive hydrogeological and environmental investigations before constructing a full-scale project (Mantoglou 2003; Zhou, et al. 2003; Abarca et al. 2006; Sutherland and Barfuss, 2011; Singh 2015; Abd-Elaty et al. 2019; Guo et al. 2019; M Armanuos et al. 2019).

Although physical and numerical models are useful in determining the optimum solutions for controlling saltwater intrusion, deficiencies in the acquisition of appropriate evidence to support the final decision are discovered. Since the scenarios of hydrogeological models for a specific aquifer cannot agree on minimizing intrusion, improving groundwater availability, being environmentally friendly, and being cost-effective. It is necessary to use decision models in conjunction with physical and numerical models to guide stakeholders toward sustainable resource management based on a set of criteria. The analytical hierarchy process (AHP) is a decision-making method that has been used alone or in conjunction with other techniques such as GIS and fuzzy logic in a variety of groundwater-related fields. Based on a broader set of criteria, this technique is used to guide stakeholders involved in groundwater development and sustainable resource management (Vaidya and Kumar 2006; Alwetaishi et al. 2017). The applications of AHP in the field of groundwater include assessing groundwater vulnerability by developing indices based on hydrogeological parameters and mapping groundwater potential zones (Arunbose et al. 2021; Osiakwan et al. 2022; Ahmadi et al. 2021; Castillo et al. 2022; Achu et al. 2020; Sajil Kumar et al. 2022; Nithya et al. 2019; Phin et al., 2022; Zghibi et al., 2020; (Mallick et al., 2019) (Shao et al., 2020). In the field of saltwater intrusion, a GIS-based AHP weighted index overlay analysis technique has been demonstrated to determine the distribution of groundwater vulnerability (Gangadharan, Nila, et al. 2016; Güllü and Kavurmacı 2023). A fuzzy-AHP evaluation model is developed for analyzing the level of seawater intrusion in long-term monitoring data from multiple river basins (Yang et al., 2022). The AHP is also used to compute weights for the GALDIT parameters, which are used to assess the vulnerability of coastal aquifers to saltwater intrusion (Pham et al., 2022).

According to the preceding overview, both traditional and artificial techniques of controlling seawater intrusion have limitations, and using physical, numerical, and decision-making models is crucial. The unconfined coastal aquifer is investigated in this study, and physical, numerical, and decision-making models are utilized to investigate surface and subsurface recharge methods, either alone or in combination with typical vertical barriers. On the other hand, the behaviors of saltwater intrusion, groundwater flow, and hydraulic head are numerically investigated using three categories of model cases:
categories (a), (b), and (c). Category (a) model cases explore the variation of hydraulic head along the aquifer in order to
determine the appropriate recharging location. The impacts of surface and subsurface recharges are explored separately or in
conjunction with a vertical barrier in categories (b) and (c). The aims of this study are: (i) to examine experimentally the
behavior of saltwater intrusion via coastal unconfined aquifers with and without vertical barrier countermeasures; (ii) to
develop a validated numerical model regarding the experimental findings of transitory saltwater intrusion; (iii) to identify the
optimal recharging location utilizing the location of the minimum hydraulic head; (v) to determine the optimal vertical barrier
depth for saltwater intrusion management; (iv) to identify the components of an effective countermeasure system, such as a
vertical barrier, surface recharge, and subsurface recharge, either alone or in combination; (vi) to develop a DMM model to
aid decision makers in the selection among several saltwater countermeasures and picking the most appropriate one depending
on various demanding scenarios.

**2 Materials and Methodologies**

Saltwater intrusion is investigated experimentally in this study by developing a laboratory physical model of an unconfined
coastal aquifer. Two experiments are carried out in this part, and dimensionless quantities are formed, namely evaluation ratios.
These evaluation ratios are used to analyze and characterize the saltwater line and hydraulic head variations of the numerical
model cases, as forthcoming later. A numerical finite difference model is created, and the validation and calibration processes
are carried out using the experimental results. Following that, numerical methods are utilized to investigate how to control
saltwater intrusion, taking into account the combined effect of using vertical barriers with surface or subsurface recharge
systems, as demonstrated by model cases divided into three categories (a, b, and c), each with seven cases. Category (a) model
cases are used to determine the location of the minimal hydraulic heads, which are suggested to be the locations of the indicated
artificial recharge systems. Categories (b) and (c) investigate the impacts of surface and subsurface recharges on saltwater
intrusion at the indicated locations, either alone or in conjunction with a vertical barrier. A classification process is then
implemented to classify model cases in each category as the best or worst model case using a developed set of ratios, namely
classification ratios. Because each model case is expected to have benefits and drawbacks, as well as several criteria governing
the model cases, the benefits and drawbacks of each model case should be quantified in order to identify the most effective
one. Following that, the most effective model case is decided on using a new DMM model based on the AHP technique. To
make the final decision, two selection levels (levels 1 and 2) are considered. Level 1 is used to select the best model case from
each category (three model cases). While level 2 is utilized for selecting the best overall model case. **Figure 1** illustrates a
flow chart for the framework of the study.

[revised manuscript text omitted]

**2.2 Evaluation Ratios**

Based on the geometry and experiment design given in the preceding section, **Figure 5** and **Table 2** list variables, parameters, and constants that affect saltwater intrusion. Following that, three dimensionless quantities are proposed for evaluating the results (see **Table 3**):

(1) Three variables, namely evaluation ratios, will be used to analyze the output results.

(2) One parameter that operates as experimental run constraints is referred to as a conditional parameter.

(3) Two geometric parameters are used to assign the hydraulic gradient and saltwater profile.

[Figure]

**Figure 5: Geometric characteristics of the experiments**

**Table 2: Definition of the geometric characteristics of the experiments**

| No. | Quantity | Type | | | Definition |
|---|---|---|---|---|---|
| | | Constant | Parameter | Variable | |
| 1 | $H_{sw}$ | √ | | | Hydraulic Head of the saltwater boundary |
| 2 | $D$ | √ | | | Sand media depth |
| 3 | $L_{media}$ | √ | | | Sand media length (experimental section length) |
| 4 | $max.L_{(in)}$ | √ | | | Maximum length of saltwater intrusion (attained for experiment 1 (base case)) |
| 5 | $D_b$ | | √ | | Vertical barrier depth |
| 6 | $X$ | | | √ | Horizontal distance from the saltwater boundary measured for any embedded point in the media sand |
| 7 | $Y$ | | | √ | Vertical distance measured from the experimental section bed for any embedded point in the media sand |
| 8 | $Y_{(sw)}$ | | | √ | Observed saltwater intrusion depth at any X distance at a specific time (t). |
| 9 | $H_h$ | | | √ | Observed hydraulic head at any X distance at a specific time (t). |
| 10 | $L_{(in)}$ | | | √ | The observed length of saltwater intrusion at a specific time (t) |

Taking into account the characteristics listed in **Table 2**, the dimensionless quantities that will be used in this study as evaluation ratios, conditional, and geometric parameters for examining the output findings are presented in **Table 3**.

**Table 3: Suggested evaluation ratios, conditional parameters and geometric parameters**

| | Quantities | Definition (Abbreviation) | Physical meaning |
|---|---|---|---|
| Evaluation Ratios | $L_{(in)}/max.L_{(in)}$ | Intrusion Ratio (IR) | Variation of intrusion length over time (t) with reference to the maximum intrusion length (base case) |
| | $Y_{(sw)}/H_{sw}$ | Salt Line Ratio (SLR) | A function demonstrates the variation in intrusion depth as a function of distance X and time (t) due to saltwater boundary head. In the comparative analysis of the results, the average SLR value (SLR$_{avg}$) will be used. |
| | $H_h/H_{sw}$ | Hydraulic Head Ratio (HHR) | A function demonstrates the variation of the hydraulic head due to the influence of the saltwater boundary head at a particular distance X and time (t). In the comparative analysis of the results, the minimum value of HHR and its location will be taken into account. |
| Conditional Parameter | $D_b/H_{sw}$ | Barrier Depth Ratio (BDR) | The ratio of barrier depth to saltwater boundary head depth. This ratio operates as an experimental run constraint. |
| Geometric Parameters | $X/L_{media}$ | Length Ratio (LR) | The horizontal distance X for a certain location in the experimental section to the length of the sand media. |
| | $Y/D$ | Depth Ratio (DR) | The vertical distance Y for a certain location in the experimental section to the total media sand depth. |

**2.3 Conceptual Model**

A proper conceptual model could be provided as a pre-set for developing a numerical model based on the experimental set and procedures previously presented in sections 2.1 and 2.2. The numerical investigation of saltwater intrusion will be conducted using either a traditional vertical barrier or artificial recharge approaches. To control seawater intrusion using a vertical barrier, various penetration depths will be simulated. Surface and subsurface recharge systems will be used as artificial recharge methods. To determine their effectiveness in controlling the saltwater intrusion problem, each of the management techniques is evaluated independently and in combination with vertical barrier. Error! Reference source not found. shows the numerical model cases being explored under different constraints. The suggested conceptual system is presented in **Figure 6**, taking into account the boundary heads, initial hydraulic grade line (HGL), barrier depth and location, and artificial recharge methods.

When there is no vertical barrier, two water zones can be identified: zone 1 (saltwater zone) and zone 2 (freshwater zone), as shown in **Figure 6a**. After using a vertical barrier, zones 1 and 2 are further partitioned into two zones: zone 1a and zone 1b for saltwater and zone 2a and zone 2b for freshwater, as shown in **Figure 6b**. The key features of the conceptual system are outlined below.

1. A constant-head saltwater boundary.

2. A time-variant head freshwater boundary that advances from the initial head to equilibrium with the saltwater boundary in the steady-state condition.

3. A vertical barrier of variable depths at a certain location.

4. A source of surface and subsurface artificial recharge.

**Table 4: The studied cases using numerical simulation**

| Category (a): using vertical barrier | |
| --- | --- |
| Model Cases | Description |
| Case1a | Base Case (Verification of experiment1) |
| Case2a | *BDR*=0.875 |
| Case3a | *BDR*=0.75 (Verification of experiment2) |
| Case4a | *BDR*=0.625 |
| Case5a | *BDR*=0.50 |
| Case6a | *BDR*=0.375 |
| Case7a | *BDR* =0.125 |
| Category (b): using vertical barrier and surface recharge | |
| Model Cases | Conditional Parameters |
| Case1b | Case1a + Surface Recharge |
| Case2b | Case2a + Surface Recharge |
| Case3b | Case3a + Surface Recharge |
| Case4b | Case4a + Surface Recharge |
| Case5b | Case5a + Surface Recharge |
| Case6b | Case6a + Surface Recharge |
| Case7b | Case7a + Surface Recharge |
| Category (c): using vertical barrier and subsurface recharge | |
| Model Cases | Conditional Parameters |
| Case1c | Case1a + borewells Recharge |
| Case2c | Case2a + borewells Recharge |
| Case3c | Case3a + borewells Recharge |
| Case4c | Case4a + borewells Recharge |
| Case5c | Case5a + borewells Recharge |
| Case6c | Case6a + borewells Recharge |
| Case7c | Case7a + borewells Recharge |

[Figure]

Figure 6: Saltwater intrusion conceptual model: (a) freshwater and saltwater zones without barrier, (b) freshwater and saltwater zones with barrier

**2.4 Numerical Model Development**

MODFLOW-2005, in conjunction with the SWI2 package, is used in this study for numerical modeling of saltwater intrusion(Harbaugh, 2005). SWI2 is a software package used to analyze three-dimensional groundwater flow, model saltwater intrusion, and calculate hydraulic heads(Bakker et al., 2013). The main advantage of using the SWI2 package is that it requires fewer cells for the simulation process than variable-density groundwater flow packages like SEAWAT. The ability of SWI2 to represent each aquifer as a single layer of cells results in significant model run-time savings.

MODPATH is a post-processing package for particle tracking that computes and displays three-dimensional pathlines based on MODFLOW output (Pollock, 2016). The MODPATH packages are used to visualize the flow behavior of both freshwater and saltwater through the sand media by visualizing the expert transport trajectories coming from the saltwater boundary, the freshwater boundary, and the flow path from the recharge area for the cases defined in Error! Reference source not found.. The particle tracking in the MODPATH package is simulated in the forward tracking direction using cylinder particle placement, as illustrated in **Figure 7b.**

On the basis of the conceptual model, the saltwater boundary cells are represented by the General-Head Boundary (GHB) package. The Time-Variant Specified-Head (CHD) package is applied to the model freshwater boundary cells to obtain the same results as the experiments, with an initial hydraulic gradient of 0.085. The recharge value for each recharge type will be relevant to the flow across the saltwater boundary for each model case b, with the constraint that the hydraulic heads do not exceed the medium sand level as a maximum value. Various discretization systems are also examined in order to provide an accurate assessment of discrepancies in head drawdowns and water balances. In this study, 8 model layers with 2320 cell discretization are used, as shown in Figure 7a. Furthermore, as shown in Figure 7b, the flow direction will be characterized as $+^{ve}Y$, $-^{ve}Y$, $+^{ve}X$, and $-^{ve}X$.

[Figure]

Figure 7: Structure of the numerical model: (a) discretization and boundary conditions, (b) particle tracking and flow directions

**2.4.1 Calibration and Verification Processes**

Many factors contribute to groundwater model inconsistency, including hydrogeological properties, discretization, potentially spatial discretization, time step, and solver parameters. Using the experimental results, many trials are carried out to calibrate the model using various hydrogeological properties, with reference to (Domenico et al. 1998; Rotz 2021). The transient stress period, on the other hand, will be assigned to be more than that needed for the experiment, with a proper equal interval time step. The impact on the heads on the cells and the accumulated volume water balance are evaluated. Following that, a verification procedure is implemented for:

1- Confirming the time when a steady-state condition occurs in based the results of experiment 1.

2- Fitting the observed saltwater line in experiments 1 and 2 for the transient and steady-state conditions.

**2.4.2 Classification Ratios**

As a starting point for selecting the best model case for controlling saltwater intrusion, four ratios are suggested to classify the model cases included in categories (a), (b), and (c). These ratios are calculated using the numerical results of the models. Each ratio is calculated for each model case and then classified by its value into best or worst. These ratios are the increase of saltwater ratio ($SLR_i$), repulsion ratio ($Rr$), wedge area ratio (WAR), and recharge ratio (RER). The four ratios are computed using the **equations 1, 2, 3,** and **4**, respectively, with the RER ratio computed only for cases in categories (b) and (c). The criteria for classifying the best model cases are that they have low values of $SLR_i$, WAR, and RER, as well as the maximum value of Rr. On the other hand, cases with high values of $SLR_i$, WAR, and RER, as well as the lowest value of Rr, are classified as the worst model cases and are not recommended for controlling saltwater intrusion. Because of the difficulty of having a model case have all the best or worst values of classification ratios to be classified as the best or worst model case (unclassified model case), it is important to use the DMM models to use the values of these classification ratios to make the final decision.

$$SLR_i = SLR_{avg}^{case\ k} - SLR_{avg}^{case1a} \tag{1}$$

$$R_r = IR_{case1a} - IR_{casek} \tag{2}$$

$$WAR = \frac{Wedge\ Area_{case(k)}}{Wedge\ Area_{case1a}} \tag{3}$$

$$RER = \frac{Recharge_{case(j)}}{Saltwater\ boundary\ Recharge_{case(j)}} \tag{4}$$

Where case(k) is any case included at any category (a, b, and c), and case(j) is the cases included at category (b) and category (c).

**2.5 Decision-Making Model (AHP technique)**

The AHP technique is commonly employed in decision-making systems designed to aid in decision-making and rate options (Saaty, 1986). Actual metrics such as pricing, headcount, or subjective opinions are used as inputs into a numerical matrix in AHP. Ratio scales and consistency indices derived from eigenvalues and eigenvectors are among the results. The AHP model is a decision-making framework that assumes decision levels have a unidirectional hierarchical relationship (Presley, 2006). AHP can study the interrelationships among all criteria using the hierarchical approach (Singh et al., 2007).

According to (Albayrak and Erensal, 2004), there are three processes that go into creating AHP: model structure (decomposition), comparative judgment of alternatives and criteria, and priority synthesis. These methods can be broken down into four stages.

In the first stage, AHP divides a complex multi-criteria decision problem into a hierarchy of interrelated elements (criteria, decision alternatives). The criteria and alternatives are arranged in a family tree-like hierarchical structure. The next stage, after the problem has been decomposed and a hierarchy has been established, is to begin the comparison judgment process to evaluate the relative importance of the criteria within the grade. The criteria are compared pairwise at each grade based on their degrees of influence and the criteria provided at the higher grade. Pairwise comparisons are based on a nine-point scale, with 1 indicating "equal importance," 3 indicating "slightly more important," 5 indicating "much more important," 7 indicating "highly more important," and 9 indicating "extremely more important" (Issa et al., 2020; Abdelwahab et al., 2021). These alternatives and criteria are evaluated based on the subjective opinions of experts represented by a point scale, including any intermediate value (2, 4, 6, and 8).

As demonstrated in Eq. (5), the result of a pairwise comparison on n criteria can be summarized in a $[X]_{(n*n)}$ evaluation matrix.

$$X = \begin{array}{cccc} C_1 & C_2 & .... & C_n \end{array}$$

$$X = \begin{bmatrix} x_{11} & x_{12} & \cdots & x_{1n} \\ x_{21} & x_{22} & \cdots & x_{2n} \\ \vdots & \vdots & \ddots & x_{3n} \\ x_{n1} & x_{n2} & \cdots & x_{nn} \end{bmatrix} \begin{array}{c} C_1 \\ C_2 \\ \\ C_n \end{array} \qquad (5)$$

Where: $c_j = 1, 2, 3\ldots n$ – the set of criteria; $x_{ij}$ ($ij = 1, 2, 3\ldots n$) – the weight quotient of the criteria; $x_{ij}=1$; $x_{ji}=1/x_{ij}$; $x_{ij}\neq 0$

The third stage, which comes after the dual comparison matrices, is to calculate the eigenvector, which shows the importance of each element in the relevant matrix with respect to the others (Albayrak and Erensal, 2004).

In Eqs (6) and (7), the % importance distribution of criterion is computed as follows:

$$b_{ij} = \frac{x_{ij}}{\sum_{i=1}^{n} x_{ij}} \qquad (6)$$

$$w_i = \frac{\sum_{j=1}^{n} b_{ij}}{n} \qquad (7)$$

Where: $b_{ij}$ – the values of the normalized matrices; $[w_i]_{n*1}$ – the percentage importance distribution of criteria; n – the number of criteria.

The fourth step is to ensure that the consistency ratio (CR) for each comparison matrix does not exceed 10% at the most.

A CR greater than 10% indicates inconsistency in the decision maker's judgments. The judgments in this case should be improved. Eqs (8) and (9) are used to compute the CR value:

$$[D_i]_{n*1} = [x_{ij}]_{n*n} * [w_i]_{n*1} \qquad (8)$$

$$\lambda_{max} = \frac{\sum_{i=1}^{n} \frac{d_i}{w_i}}{n} \qquad (9)$$

Where: $\lambda_{max}$ is the matrix's largest eigenvector and $[D_i]_{n*1}$ is the weighted matrix.

Random Index (RI) is another value required to calculate CR. (Özat, 2013) provides the data, which includes the RI values, which are constant numbers determined by the N value. Eq. (10) specifies the calculation of the CR value based on this information.

$$CR = \frac{\lambda_{max}-n}{(n-1)*RI} \qquad (10)$$

Where CR is the consistency ratio, $\lambda_{max}$ is the matrix's largest eigenvector, RI is the random index, and n is the number of criteria.

In this study, it is suggested that an AHP-based model be used on two levels to find the best model case by comparing these model cases with the help of many ratios as a selection criterion. Through the AHP analysis, the model cases will be named as alternatives. The three alternatives (cases 1a, 1b, and 1c) with no vertical barrier countermeasure will be eliminated from the total number of alternatives, 21 alternatives, reducing the total number of alternatives to 18 alternatives (six cases in each category). Level (1) involves the model dealing with three categories (a, b, and c) in order to select the best alternative from each. There are four criteria in category (a) (Rr, SLR$_i$, Minimum HHR, and WAR), and five in categories (b) and (c), with RER which is indicating the artificial recharge, which is exclusively utilized in categories (b) and (c). The top three alternatives from each of the three categories that emerged from level (1) can be used to create the final choice for the best alternative at level (2). Pairwise comparisons with other criteria aid in determining the relative importance of each criterion in the hierarchical structuring of the problem. The model's first level consists of one matrix (4x4) for category (a) alternatives and one matrix (5x5) for categories (b) and (c) alternatives, reflecting the relative weights of the criteria as outputs. Moreover, five matrices (6x6) show the relative weight among the alternatives in the case of each criterion. The model, on the other hand, takes the same matrix for criteria weights and five matrices, each of which is (3x3), and expresses the relative weight among the final three alternatives for each criterion in its second level.

**3 Results and Discussions**

**3.1 Calibration and Verification of the Numerical Model**

The steady-state condition in experiment 1 (the base case) occurs 90 minutes after the experiment begins. As a validation of the numerical model steady-state simulation, **Figure 8** shows the observed and simulated saltwater lines for various simulation times greater than 90 minutes. The figure shows that the simulated saltwater line closely matches the observed one, with RMSE values ranging from 0.90 to 1.19 for time ranging from 90 to 120 minutes which confirm the occurrence of a steady-state condition after 90 minutes.

[Figure]

**Figure 8: Observed and simulated saltwater lines for experiment 1 (Case1a) under steady-state conditions at intervals longer than 90 minutes.**

For transient results, the saltwater line for experiments 1 and 2 for simulation times of 30, 60, and 90 minutes is used to verify the corresponding results of the numerical model, as shown in **Figure 9** and **Figure 10**. Both figures show that the model produces reasonable simulated results for the saltwater lines (case3a) when compared to the observed ones. **Table 5** also shows the calibrated hydrogeological properties of the verified numerical model, including hydraulic conductivities in X, Y, and Z directions ($k_x$, $k_y$, $k_z$), specific yield ($S_y$), specific storage ($S_s$), and effective porosity ($\eta$). The upcoming analysis will consider the results at 90 minutes as a steady-state condition.

[Figure]

**Figure 9: Observed and simulated saltwater lines for experiment1 (Case1a) for transient state condition: (a) 30min, (b) 60min, (c) 90min**

[Figure]

**Figure 10: Observed and simulated saltwater lines for experiment2 (Case3a) for transient state condition: (a) 30min, (b) 60min, (c)**
**90min**

**Table 5: Calibrated values of the hydrogeological properties**

| Hydrogeological Properties | $k_x(cm/s)$ | $k_y(cm/s)$ | $k_z(cm/s)$ | $S_y$ | $S_s$ | $\eta$ |
|---|---|---|---|---|---|---|
| Values | 0.0069 | 0.0069 | 0.03 | 0.04 | 0.0619 | 0.0428 |

**3.2 Behavior evaluation of saltwater intrusion, flow, and hydraulic heads for categories (a), (b), and (c) model cases**

**3.2.1 Saltwater intrusion and flow behaviors in category(a) model cases**

The modeling results of saltwater intrusion and the accompanying flow behavior for the cases in category (a) will be discussed
in this section. Two evaluation ratios are considered, including IR and the $SLR_{avg}$. Moreover, conditional parameters (BDR)
and geometrical parameters (LR and DR) will be considered through the discussion. **Figure 11** depicts these outcomes, and
**Table 6** provides a summary of the results.
Figures from **Figure 11a1** to **Figure 11g1** as well as
**Table** 6 reveal that case 2a, which uses a vertical barrier with high BDR values, has the lowest evaluation ratio values (see
**Figure 11b1**). Case 7a's evaluation ratios, on the other hand, have the highest values when a vertical barrier with low BDR
values is applied (see **Figure 11g**1). Given these findings, flow behavior through the media sand needs to be investigated as
an explanation for the variation in evaluation ratios.
Figures from **Figure 11a2** to **Figure 11g2** depict the flow behavior of freshwater and saltwater. **Figure 11a2** depicts the flow
behavior of case1a, demonstrating that the flow in zone1 takes two directional flows: $+^{ve}$ Y and $+^{ve}$X. The $+^{ve}$Y flow conserves
hydraulic heads near the saltwater boundary at the media sand level. Furthermore, the $+^{ve}$X flow forces freshwater above the
saltwater line to flow in the same direction as the saltwater. Freshwater flow directions in zone 2 are $-^{ve}$X and$+^{ve}$Y in the upper
half of the zone and $-^{ve}$X and $-^{ve}$Y in the lower half of the zone. Because of the $+^{ve}$Y and $-^{ve}$Y flows in zone 2, a separation line
with a DR value in the range of 0.37 to 0.45 could be identified, as illustrated in
**Table** 6 and shown in **Figure 11a2**. Along zone2, the $+^{ve}$Y flow direction conserves hydraulic head. In the upcoming analysis,
the DR value of the separation line will be termed $DR_{separation}$.

**Figure 11b2** shows that the vertical barrier impedes freshwater flows from zone2a to zone2b, creating overlaying pressure in zone1a, resulting in a dramatic drop and rise of the saltwater line shortly before and after the vertical barrier. Moreover, as shown in

**Table** 6, the value of DR$_{separation}$ increases to be in the range of 0.40 to 0.50 when compared to case1a.

The flow of freshwater from zone2a to zone2b is boosted by continuing to decrease BDR values, producing fluctuations in the saltwater line. Because of this flow, the overlying pressure of freshwater on zone 1a is reduced, leading the SLR value in this zone to rise (see **Figures 11c2, 11d2, 11e2, 11f2, 11g2**). Furthermore, these figures and

**Table** 6 show that DR$_{separation}$ values are increasing, indicating that the majority of the freshwater flow in zone 2b is in the -$^{ve}$Y

flow direction, resulting in hydraulic head reduction through this zone.

Based on the given results, it is possible to conclude that case2a has the lowest evaluation ratio values among the other cases.

Furthermore, large DR$_{separation}$ values, such as case7a, limit the hydraulic heads, creating an excess increase in the evaluation ratios (see **Figure 11g1 and Figure 11g2**). In addition, adopting a vertical barrier with a high BDR ratio could effectively manage the saltwater intrusion. Furthermore, management of saltwater intrusion will be considered in this study by managing the DR$_{separation}$ as well as the hydraulic heads along zone2b using groundwater artificial recharge in conjunction with the use of a vertical barrier.

[Figure]

                                                  384

[Figure]

**Figure 11: Simulated saltwater lines and groundwater flow behavior of the category (a) model cases: (a) case1a, (b) case2a, (c) case**

**3a, (d) case4a, (e) case5a, (f) case6a, (g) case7a**

**Table 6: Values of the evaluation ratios and DR values for category (a) model cases**

| Cases | Conditional Parameters | Evaluation Ratios | | Geometrical Parameters | |
|---|---|---|---|---|---|
| | BDR | IR | $SLR_{avg}$ | $LR_{Intrusion}$ | $DR_{separation}$ |
| Case1a | --- | 0.97 | 0.28 | 0.45 | 0.37-0.45 |
| Case2a | 0.875 | 0.83 | 0.20 | 0.39 | 0.40-0.50 |
| Case3a | 0.75 | 0.90 | 0.23 | 0.42 | 0.50-0.68 |
| Case4a | 0.625 | 0.97 | 0.25 | 0.45 | 0.60-0.70 |
| Case5a | 0.50 | 0.97 | 0.31 | 0.45 | 0.69-0.75 |
| Case6a | 0.375 | 1.05 | 0.29 | 0.48 | 0.71-0.78 |
| Case7a | 0.125 | 1.05 | 0.32 | 0.48 | 0.76-0.85 |

**3.2.2 Hydraulic head variations in category(a) model cases**

As illustrated in **Figure 12**, the hydraulic head variations indicated by the HHR evaluation ratio are investigated for the category(a) model cases. This figure illustrates the relationship between HHR and LR ratios by displaying the minimum HHR

values and their locations along the aquifer. **Figure 12** shows that the hydraulic head of case7a has the lowest HHR value of

0.91 compared with the other cases (cases 1a-6a) located at a LR value of 0.55 (see **Figure 12g**). On the other hand, Case1a, has the highest value of the minimum HHR (0.98), and a location has a LR of 0.44, as shown in **Figure 12a**.

[Figure]

[Figure]

**Figure 12: Values and locations of the minimum HHR of the category (a) model cases: (a) case1a, (b) case2a, (c) case 3a, (d) case4a, (e) case5a, (f) case6a, (g) case7a**

The above results can be summarized as illustrated in **Figure 13**, which depicts the effect of BDR on the location (LR), the value of the minimum HHR, and the IR ratios. The minimum hydraulic head is located at zone2b for all study cases (case1a-case7a), with LR values ranging from 0.45 to 0.55 and corresponding minimum HHR values ranging from 0.91 to 0.98. On the other hand, the maximum IR occurs for both cases 6a and 7a with a value of 1.05 when using a BDR in the value range from 0.25 to 0.38. Given these findings, increasing the hydraulic head represented by HHR could effectively control saltwater intrusion when combined with the vertical barrier countermeasure. For this purpose, using groundwater artificial recharge, whether by surface or subsurface recharge, at the location of the minimum HHR value (LR in the range from 0.45 to 0.55), combined with the use of a vertical barrier, could be used to control saltwater intrusion, as will be discussed in the following sections of this study.

[Figure]

**Figure 13: Effect of BDR on the IR and minimum HHR values and locations**

**3.2.3 Saltwater intrusion and flow behaviors in categories (b) and (c) model cases**

Groundwater artificial recharge is used to control saltwater intrusion in zone2b along the LR range (from 0.45 to 0.55), which has a minimum value of HHR for preserving its value at the unity value. Surface and subsurface recharge are numerically discussed, either separately or in conjunction with the vertical barrier, as shown in Error! Reference source not found. for category (b) and category (c) model cases. The recharge is applied along the whole range of LR values from 0.45 to 0.55 for surface recharge. In contrast, for subsurface recharge, the recharge is applied as a line of wells at the midpoint of the same LR

range at a value of 0.5. The results of category (b) and category (c) study cases will be compared with the base case results (case1a) and the corresponding cases of category (a) in the following discussions, as depicted in **Figure 14** and summarized in **Table 8**.

As an analysis of the saltwater intrusion based on the evaluation ratios from **Figure 14a1** to **Figure 14g1** as well as **Table 8**, it is found that case3b has the lowest IR value among all the model cases included in categories (a), (b), and (c). However, the

$SLR_{avg}$ values of case2a and case3b are the lowest, with case2a having a lower value than case3b.

The saltwater and freshwater flow behaviors could be described from **Figure 14a2** to **Figure 14g2**. In case1b, the surface recharge works as a hydraulic barrier that prevents saltwater from flowing in the $^{+ve}X$ direction as well as forces it to flow intensively in the $^{+ve}Y$ direction. This behavior causes an increase in the $SLR_{avg}$, compared with that of case1a, and the majority of recharged freshwater is forced to take a $^{+ve}X$ direction (see **Figure 14a2**). The flow behavior in case1c is similar to that in case1b (see **Figure 14a3**), but its $SLR_{avg}$ is higher, indicating that the countermeasure effect of subsurface recharging, which is a line of wells, is less than that of surface recharge, which is a water mass.

In contrast to Case 2b, the value of SLR rises due to the $^{-ve}X$ flow direction of surface recharge towards the neck area beneath the vertical barrier, preventing the saltwater line from intruding (**Figure 14b2**). Because well recharge has a lower effect than surface recharge, the IR and LR ratios are higher in case 2c than in case 2b, as shown in **Figures 14b2** and **14b3** and illustrated in **Table 8**.

In the case3b flow behavior, freshwater flows intensively from zone 2a to zone 2b (see **Figure 14c2**), causing $SLR_{avg}$ to decline to become the least among the category (b) model cases. Because of the poor influence of well recharge, the $SLR_{avg}$ value for case3c is greater than that of case3b, as demonstrated in **Table 8**. By continuing to lower BDR for cases 4b, 5b, 6b, and 7b, as well as the corresponding cases in category (c), the freshwater flows from zone 2a to zone 2b in the $^{+ve}X$ direction, which reduces the effect of surface and well recharge, as shown in **Table 8**.

Because of the artificial recharge that applies in categories (b) and (c), the hydraulic heads along the experiment section are unchanged for all model cases, and the $DR_{separation}$ is nearly the same with a value range from 0.75 to 0.90.

Based on the findings, it is possible to conclude that artificial aquifer recharging along the LR values from 0.45 to 0.55, which has a minimum value of the HHR ratio to conserve its value, as well as the unity accompanied by using a vertical barrier, has a significant effect on controlling saltwater intrusion. Furthermore, because of its body mass, surface recharge is more efficient than well recharge. Conclusively, the value of IR, as an evaluation ratio, for case3b is the lowest among the whole cases included in categories (a), (b), and (c). However, the minimum value of $SLR_{avg}$ is achieved in case2a, confirming the efficient combination of the vertical barrier and surface recharge at the location of the minimum HHR (LR in the range from 0.45 to

0.55).

[Figure]

[Figure]

[Figure]

**Figure 14: Simulated saltwater lines and groundwater flow behavior of the category (b) and (c) model cases: (a) case1b &1c, (b)**
**case2b&2c, (c) case 3b&3c, (d) case4b&4c, (e) case5b&5c, (f) case6b&6c, (g) case7b&7c**

**Table 7: Values of the evaluation ratios and DR values for category (a) model cases**

| Category | Cases | Conditional Parameters | Evaluation Ratios | | Geometrical Parameters | |
|---|---|---|---|---|---|---|
| | | BDR | IR | $SLR_{avg}$ | $LR_{Intrusion}$ | $DR_{separation}$ |
| Category(a) | Case1a | --- | 0.97 | 0.28 | 0.45 | 0.37-0.45 |
| | Case2a | 0.875 | 0.83 | 0.20 | 0.39 | 0.40-0.50 |
| | Case3a | 0.75 | 0.90 | 0.23 | 0.42 | 0.50-0.68 |
| | Case4a | 0.625 | 0.97 | 0.25 | 0.45 | 0.60-0.70 |
| | Case5a | 0.50 | 0.97 | 0.31 | 0.45 | 0.69-0.75 |
| | Case6a | 0.375 | 1.05 | 0.29 | 0.48 | 0.71-0.78 |
| | Case7a | 0.125 | 1.05 | 0.32 | 0.48 | 0.76-0.85 |
| Category(b) | Case1b | --- | 1.0 | 0.39 | 0.47 | 0.80-0.85 |
| | Case2b | 0.875 | 0.75 | 0.40 | 0.35 | 0.80-0.85 |
| | Case3b | 0.75 | 0.68 | 0.34 | 0.32 | 0.80-0.90 |
| | Case4b | 0.625 | 0.81 | 0.44 | 0.38 | 0.80-0.90 |
| | Case5b | 0.50 | 0.81 | 0.51 | 0.38 | 0.75-0.80 |
| | Case6b | 0.375 | 0.81 | 0.54 | 0.38 | 0.75-0.80 |
| | Case7b | 0.125 | 0.81 | 0.37 | 0.38 | 0.80-0.90 |
| Category(c) | Case1c | --- | 1.05 | 0.41 | 0.49 | 0.80-0.85 |
| | Case2c | 0.875 | 0.82 | 0.35 | 0.38 | 0.80-0.85 |
| | Case3c | 0.75 | 0.82 | 0.38 | 0.38 | 0.80-0.90 |
| | Case4c | 0.625 | 0.85 | 0.45 | 0.40 | 0.80-0.90 |
| | Case5c | 0.50 | 0.85 | 0.52 | 0.40 | 0.75-0.80 |
| | Case6c | 0.375 | 0.85 | 0.57 | 0.40 | 0.75-0.80 |
| | Case7c | 0.125 | 0.75 | 0.39 | 0.40 | 0.80-0.90 |

**3.2.4 Hydraulic head variations in categories (b) and (c) model cases**

**Figure** from **15a** to **15g** depict the hydraulic heads along the aquifer as represented by the HHR ratio for cases in categories (b) and (c) compared to category (a). The hydraulic heads for all cases have been conserved along the LR ratio from 0.45 to 0.55, which has the minimum value of HHR to have the unity value, and the losses through the vertical barrier are greatly reduced when compared to those of category (a).

[Figure]

**Figure 15: Hydraulic head variation along the aquifer for categories (b) and (c) compared with those of category (a) model cases:**
**(a) case1a&1b&1c, (b) case2b&2b&2c, (c) case 3b&3b&3c, (d) case4b&4b&4c, (e) case5b&5b&5c, (f) case6b&6b&6c, (g)**
**case7b&7b&7c**

**3.3 Classification of model cases**

The classification ratios described in Section 2.3.2, Classification Ratios, are summarized and classified in **Figure 16** and

**Table 8**. **Figure 16a** presents the SLR$_{avg}$ and Rr values for each model case, whereas **Figure 16b** depicts the WAR and RER

values. **Figures 16** and **Table 8** show that case3b has the best Rr value of 0.29. Case2a, on the other hand, has the best SLR$_{avg}$

and WAR values of -0.08 and 0.76, respectively. Furthermore, case7c has the best RER value of 1.91. On the contrary, case1c has the worst Rr and WAR values of -0.07 and 2.18, respectively. Furthermore, case6b has the worst RER value of 3.62.

Moreover, case6c has the worst SLR$_{avg}$ value of 3.62. The remaining model cases are categorized as unclassified model cases.

Based on the findings, it is difficult to determine which model case is the most successful scenario to implement as a saltwater intrusion countermeasure. As a result, the DMM model is needed for determining the most effective model case.

[Figure]

(a)

(b)

**Figure 16: Classifications of model cases included in categories (a), (b), and (c): (a) SLR$_{avg}$ and R$_r$ values, (b) WAR and RER values**

**Table 8: Model cases classification according to values of classification ratios**

| Classification Ratio | Best | Worst |
|---|---|---|
| SLR$_i$ | Case2a (-0.08) | Case6c (0.29) |
| Rr | Case3b (0.29) | Case1c (-0.07) |
| WAR | Case2a (0.76) | Case1c (2.18) |
| RER | Case7c (1.91) | Case6b (3.62) |

**3.4 Selecting the Most Effective Model Case (AHP application results)**

As previously stated, the AHP model is applied to the numerical model results at two different levels of selection (levels (1)

and (2)). Model cases are referred to as alternatives at this stage, and selected ratios among the evaluation and classification ratios are referred to as criteria. Level (1) needs to determine the best alternative in each category. Furthermore, level (2) is for deciding the best alternative. For category (a) alternatives, the criteria values used include SLR$_i$, minimum HHR, Rr, and

WAR, and the RER is added over these ratios for categories (b) and (c).

**3.4.1 Level (1) Results**

For the alternatives in each category, the model is applied at level (1) using the weights chosen among criteria as shown in

**Figure 17**. According to **Figure 17a**, minimum HHR has the largest weight for category (a) alternatives, followed by Rr. Also,

WAR has the lowest weight. Similarly, for category (b) alternatives (see **Figure 17b**), the same rating is observed for minimum

HHR (the highest weight), followed by Rr, while WAR has the lowest weight.

[Figure]

**Figure 17: Level (1) criteria relative weights for different categories: (a) category (a), (b) category (b), and (c)**

**Figure 18** illustrates the results of the relative weights in the three categories for each alternative. It is clear that case 2a ranks first in this category with a relative weight of 38.72%, then case 3a, and case 6a comes in last (
[revised manuscript text omitted]

Albayrak, E. and Erensal, Y. C.: Using analytic hierarchy process (AHP) to improve human performance: An application of multiple criteria decision making problem, J. Intell. Manuf., 15, 491–503, https://doi.org/10.1023/B:JIMS.0000034112.00652.4c, 2004.

Alwetaishi, M., Gadi, M., and Issa, U. H.: Reliance of building energy in various climatic regions using multi criteria, Int. J. Sustain. Built Environ., 6, 555–564, https://doi.org/10.1016/j.ijsbe.2017.12.002, 2017.

Anders, R., Mendez, G. O., Futa, K., and Danskin, W. R.: A geochemical approach to determine sources and movement of saline groundwater in a coastal aquifer., Ground Water, 52, 756–768, https://doi.org/10.1111/gwat.12108, 2014.

Arunbose, S., Srinivas, Y., Rajkumar, S., Nair, N. C., and Kaliraj, S.: Remote sensing, GIS and AHP techniques based investigation of groundwater potential zones in the Karumeniyar river basin, Tamil Nadu, southern India, Groundw. Sustain. Dev., 14, 100586, https://doi.org/https://doi.org/10.1016/j.gsd.2021.100586, 2021.

ASCE: Standard Guidelines for Artificial Recharge of Ground Water, EWRI/ASCE., American Society of Civil Engineers, https://doi.org/10.1061/9780784405482, 2001.

Bakker, M., Schaars, F., Hughes, J. D., Christian D. Langevin, A., and Dausman, A. M.: Documentation of the seawater intrusion (SWI2) package for MODFLOW: U.S. Geological Survey Techniques and Methods, book 6, chap. A46, 47 p. pp., 2013.

Cai, J., Taute, T., and Schneider, M.: Recommendations of Controlling Saltwater Intrusion in an Inland Aquifer for Drinking-Water Supply at a Certain Waterworks Site in Berlin (Germany), Water Resour. Manag., 29, 2221–2232, https://doi.org/10.1007/s11269-015-0937-7, 2015.

Cary, L., Petelet-Giraud, E., Bertrand, G., Kloppmann, W., Aquilina, L., Martins, V., Hirata, R., Montenegro, S., Pauwels, H., Chatton, E., Franzen, M., Aurouet, A., Lasseur, E., Picot, G., Guerrot, C., Fléhoc, C., Labasque, T., Santos, J. G., Paiva, A., Braibant, G., and Pierre, D.: Origins and processes of groundwater salinization in the urban coastal aquifers of Recife (Pernambuco, Brazil): A multi-isotope approach, Sci. Total Environ., 530–531, 411–429, https://doi.org/https://doi.org/10.1016/j.scitotenv.2015.05.015, 2015.

Castillo, J. L., Martínez Cruz, D. A., Ramos Leal, J. A., Tuxpan Vargas, J., Rodríguez Tapia, S. A., and Marín Celestino, A. E.: Delineation of Groundwater Potential Zones (GWPZs) in a Semi-Arid Basin through Remote Sensing, GIS, and AHP Approaches, https://doi.org/10.3390/w14132138, 2022.

Domenico, P. A., Schwartz, F. W., and SCHWARTZ, F. A.: Physical and Chemical Hydrogeology, Wiley, 1998.

Eissa, M. A., de Dreuzy, J.-R., and Parker, B.: Integrative management of saltwater intrusion in poorly-constrained semi-arid coastal aquifer at Ras El-Hekma, Northwestern Coast, Egypt, Groundw. Sustain. Dev., 6, 57–70, https://doi.org/https://doi.org/10.1016/j.gsd.2017.10.002, 2018.

G.U.N.T: https://www.gunt.de/en/products/hydraulics-for-civil-engineering/hydraulic-engineering/seepage-flow/visualisation-of-seepage-flows/070.16900/hm169/glct-1:pa-148:ca-181:pr-768, last access: 14 September 2023.

Gangadharan, R., Nila, R., and Vinoth, S.: Assessment of groundwater vulnerability mapping using AHP method in coastal watershed of shrimp farming area, Arab. J. Geosci., 9, 1–14, https://doi.org/10.1007/s12517-015-2230-8, 2016.

Güllü, Ö. and Kavurmacı, M.: Investigation of temporal variation of groundwater salinity potential using AHP-based index, Environ. Monit. Assess., 195, 365, https://doi.org/10.1007/s10661-023-10993-5, 2023.

Harbaugh, A. W.: MODFLOW-2005: the U.S. Geological Survey modular ground-water model--the ground-water flow process, Techniques and Methods, https://doi.org/10.3133/tm6A16, 2005.

Hasan, M. B., Driessen, P. P. J., Majumder, S., Zoomers, A., and van Laerhoven, F.: Factors Affecting Consumption of Water from a Newly Introduced Safe Drinking Water System: The Case of Managed Aquifer Recharge (MAR) Systems in Bangladesh, https://doi.org/10.3390/w11122459, 2019.

Huang, P.-S. and Chiu, Y.-C.: A Simulation-Optimization Model for Seawater Intrusion Management at Pingtung Coastal

Area, Taiwan, Water, 10, https://doi.org/10.3390/w10030251, 2018.

Issa, U. H., Mosaad, S. A. A., and Salah Hassan, M.: Evaluation and selection of construction projects based on risk analysis,
Structures, 27, 361–370, https://doi.org/https://doi.org/10.1016/j.istruc.2020.05.049, 2020.

Kallioras, A., Pliakas, F.-K., Schüth, C., and Rausch, R.: Methods to Countermeasure the Intrusion of Seawater into Coastal
Aquifer Systems, in: Wastewater Reuse and Management, 470–490, https://doi.org/10.1007/978-94-007-4942-9_17, 2013.

M Armanuos, A., Gamal Eldin Ibrahim, M., Mahmod, W., Takemura, J., and Yoshimura, C.: Analysing the Combined Effect
of Barrier Wall and Freshwater Injection Countermeasures on Controlling Saltwater Intrusion in Unconfined Coastal Aquifer
Systems, Water Resour. Manag., 33, https://doi.org/10.1007/s11269-019-2184-9, 2019.

Maliva, R. G.: ASR and Aquifer Recharge Using Wells, in: Anthropogenic Aquifer Recharge: WSP Methods in Water
Resources Evaluation Series No. 5, Springer International Publishing, Cham, 381–436, https://doi.org/10.1007/978-3-030-
11084-0_13, 2020a.

Maliva, R. G.: Surface-Spreading AAR Systems (Non-basin), in: Anthropogenic Aquifer Recharge: WSP Methods in Water
Resources Evaluation Series No. 5, Springer International Publishing, Cham, 517–565, https://doi.org/10.1007/978-3-030-
11084-0_16, 2020b.

Mallick, J., Khan, R. A., Ahmed, M., Alqadhi, S. D., Alsubih, M., Falqi, I., and Hasan, M. A.: Modeling Groundwater Potential
Zone in a Semi-Arid Region of Aseer Using Fuzzy-AHP and Geoinformation Techniques, https://doi.org/10.3390/w11122656,
2019.

Mantoglou, A.: Pumping management of coastal aquifers using analytical models of saltwater intrusion, Water Resour. Res.,
39, https://doi.org/https://doi.org/10.1029/2002WR001891, 2003.

Nithya, C. N., Srinivas, Y., Magesh, N. S., and Kaliraj, S.: Assessment of groundwater potential zones in Chittar basin,
Southern India using GIS based AHP technique, Remote Sens. Appl. Soc. Environ., 15, 100248,
https://doi.org/https://doi.org/10.1016/j.rsase.2019.100248, 2019.

Osiakwan, G. M., Gibrilla, A., Kabo-Bah, A. T., Appiah-Adjei, E. K., and Anornu, G.: Delineation of groundwater potential
zones in the Central Region of Ghana using GIS and fuzzy analytic hierarchy process, Model. Earth Syst. Environ., 8, 5305–
5326, https://doi.org/10.1007/s40808-022-01380-z, 2022.

Özat, S. T.: Determination of criterion that affects supplier selection in public administration software tenders and selection of
supplier2013 ,.

Panthi, J., Pradhanang, S. M., Nolte, A., and Boving, T. B.: Saltwater intrusion into coastal aquifers in the contiguous United
States — A systematic review of investigation approaches and monitoring networks, Sci. Total Environ., 836, 155641,
https://doi.org/https://doi.org/10.1016/j.scitotenv.2022.155641, 2022.

Pham, Q. N., Ta, T. T., Le Tran, T., Pham, T. T., and Nguyen, T. C.: Assessment of Saltwater Intrusion Vulnerability in the
Coastal Aquifers in Ninh Thuan, Vietnam BT  - Global Changes and Sustainable Development in Asian Emerging Market
Economies Vol. 2: Proceedings of EDESUS 2019, edited by: Nguyen, A. T. and Hens, L., Springer International Publishing,
Cham, 703–712, https://doi.org/10.1007/978-3-030-81443-4_45, 2022.

Phin, T. T., Hoa, D. T. B., Trong, T. D., Hai, D. T., and Que, P. T. N.: Mapping vulnerability water supply in Rach Gia city
due to saline intrusion on using analytical hierarchy process, Sustain. Water Resour. Manag., 8, 137,
https://doi.org/10.1007/s40899-022-00712-2, 2022.

Pollock, D. W.: User guide for MODPATH Version 7—A particle-tracking model for MODFLOW, Open-File Report, Reston,
VA, 41 pp., https://doi.org/10.3133/ofr20161086, 2016.

Pramada, S. K., Minnu, K. P., and Roshni, T.: Insight into sea water intrusion due to pumping: a case study of Ernakulam
coast, India, ISH J. Hydraul. Eng., 27, 442–451, https://doi.org/10.1080/09715010.2018.1553642, 2021.

Presley, A.: ERP investment analysis using the strategic alignment model, Manag. Res. News, 29, 273–284, 2006.

Qi, S.-Z. and Qiu, Q.-L.: Environmental hazard from saltwater intrusion in the Laizhou Gulf, Shandong Province of China,
Nat. Hazards, 56, 563–566, https://doi.org/10.1007/s11069-010-9686-3, 2011.

Raja Shekar, P. and Mathew, A.: Assessing groundwater potential zones and artificial recharge sites in the monsoon-fed
Murredu river basin, India: An integrated approach using GIS, AHP, and Fuzzy-AHP, Groundw. Sustain. Dev., 23, 100994,
https://doi.org/https://doi.org/10.1016/j.gsd.2023.100994, 2023.

Ríos, I. H., Cruz-Pérez, N., Chirivella-Guerra, J. I., García-Gil, A., Rodríguez-Alcántara, J. S., Rodríguez-Martín, J.,
Marazuela, M. Á., and Santamarta, J. C.: Proposed recharge of island aquifer by deep wells with regenerated water in Gran
Canaria (Spain), Groundw. Sustain. Dev., 22, 100959, https://doi.org/https://doi.org/10.1016/j.gsd.2023.100959, 2023.

Robinson, G., Ahmed, A. A., and Hamill, G. A.: Experimental saltwater intrusion in coastal aquifers using automated image
analysis: Applications to homogeneous aquifers, J. Hydrol., 538, 304–313,
https://doi.org/https://doi.org/10.1016/j.jhydrol.2016.04.017, 2016.

Rotz, R.: Hydrogeologic Properties of Earth Materials and Principles of Groundwater Flow, Groundwater, 59,
https://doi.org/10.1111/gwat.13085, 2021.

Saaty, T. L.: Axiomatic foundation of the analytic hierarchy process, Manag. Sci., 32, 841–855., 1986.

Sajil Kumar, P. J., Elango, L., and Schneider, M.: GIS and AHP Based Groundwater Potential Zones Delineation in Chennai
River Basin (CRB), India, Sustainability, 14, https://doi.org/10.3390/su14031830, 2022.

Salehi Shafa, N., Babazadeh, H., Aghayari, F., and Saremi, A.: Optimal utilization of groundwater resources and artificial
recharge system of Shahriar plain aquifer, Iran, Phys. Chem. Earth, Parts A/B/C, 129, 103358,
https://doi.org/https://doi.org/10.1016/j.pce.2023.103358, 2023.

Shao, Z., Huq, M. E., Cai, B., Altan, O., and Li, Y.: Integrated remote sensing and GIS approach using Fuzzy-AHP to delineate
and identify groundwater potential zones in semi-arid Shanxi Province, China, Environ. Model. Softw., 134, 104868,
https://doi.org/https://doi.org/10.1016/j.envsoft.2020.104868, 2020.

Shi, L. and Jiao, J. J.: Seawater intrusion and coastal aquifer management in China: a review, Environ. Earth Sci., 72, 2811–
2819, https://doi.org/10.1007/s12665-014-3186-9, 2014.

Singh, Murty, H. R., Gupta, S. K., and Dikshit, A. K.: Development of composite sustainability performance index for steel
industry, Ecol. Indic., 7, 565–588, https://doi.org/https://doi.org/10.1016/j.ecolind.2006.06.004, 2007.

Singh, A.: Managing the environmental problem of seawater intrusion in coastal aquifers through simulation–optimization
modeling, Ecol. Indic., 48, 498–504, https://doi.org/https://doi.org/10.1016/j.ecolind.2014.09.011, 2015.

Srinivasamoorthy, S. G. A.-S. G. A.-K.: Application of Geophysical and Hydrogeochemical Tracers to Investigate Salinisation
Sources in Nagapatinam and Karaikal Coastal Aquifers, South India, Aquat. Procedia, v. 4, 65-71–2015 v.4,
https://doi.org/10.1016/j.aqpro.2015.02.010, 2015.

Sutar, A. and Rotte, V.: Prevention of Saltwater Intrusion: A Laboratory-Scale Study on Electrokinetic Remediation, 389–400,
https://doi.org/10.1007/978-981-16-5501-2_31, 2022.

Vaidya, O. S. and Kumar, S.: Analytic hierarchy process: An overview of applications, Eur. J. Oper. Res., 169, 1–29,
https://doi.org/https://doi.org/10.1016/j.ejor.2004.04.028, 2006.

Wadi, D., Wu, W., Malik, I., Fuad, A., and Thaw, M. M.: Assessment and feasibility of the potential artificial groundwater
recharge in semi-arid crystalline rocks context, Biteira district, Sudan, Sci. African, 17, e01298,
https://doi.org/https://doi.org/10.1016/j.sciaf.2022.e01298, 2022.

Yang, H., Jia, C., Li, X., Yang, F., Wang, C., and Yang, X.: Evaluation of seawater intrusion and water quality prediction in
Dagu River of North China based on fuzzy analytic hierarchy process exponential smoothing method, Environ. Sci. Pollut.
Res., 29, 66160–66176, https://doi.org/10.1007/s11356-022-19871-y, 2022.

Zghibi, A., Mirchi, A., Msaddek, M. H., Merzougui, A., Zouhri, L., Taupin, J.-D., Chekirbane, A., Chenini, I., and Tarhouni,
J.: Using Analytical Hierarchy Process and Multi-Influencing Factors to Map Groundwater Recharge Zones in a Semi-Arid
Mediterranean Coastal Aquifer, Water, 12, https://doi.org/10.3390/w12092525, 2020.

Zhou, X., Chen, M., and Liang, C.: Optimal schemes of groundwater exploitation for prevention of seawater intrusion in the
Leizhou Peninsula in southern China, Environ. Geol., 43, 978–985, https://doi.org/10.1007/s00254-002-0722-9, 2003.